# Adolescent neurostimulation of dopamine circuit reverses genetic deficits in frontal cortex function

**Surjeet Mastwal[1†], Xinjian Li[1†], Rianne Stowell[2†], Matthew Manion[1], Wenyu Zhang[1,2], Nam-Shik Kim[3], Ki-Jun Yoon[3], Hongjun Song[3], Guo-Li Ming[3], Kuan Hong Wang[1,2]***

[1]Unit on Neural Circuits and Adaptive Behaviors, National Institute of Mental Health, Bethesda, United States; [2]Department of Neuroscience, Del Monte Institute for Neuroscience, University of Rochester Medical Center, Rochester, United States; [3]Department of Neuroscience, Mahoney Institute for Neurosciences, Perelman School of Medicine, University of Pennsylvania, Philadelphia, United States

**Abstract** Dopamine system dysfunction is implicated in adolescent-onset neuropsychiatric disorders. Although psychosis symptoms can be alleviated by antipsychotics, cognitive symptoms remain unresponsive and novel paradigms investigating the circuit substrates underlying cognitive deficits are critically needed. The frontal cortex and its dopaminergic input from the midbrain are implicated in cognitive functions and undergo maturational changes during adolescence. Here, we used mice carrying mutations in *Arc* or *Disc1* to model mesofrontal dopamine circuit deficiencies and test circuit-based neurostimulation strategies to restore cognitive functions. We found that in a memory-guided spatial navigation task, frontal cortical neurons were activated coordinately at the decision-making point in wild-type but not *Arc*-/- mice. Chemogenetic stimulation of midbrain dopamine neurons or optogenetic stimulation of frontal cortical dopamine axons in a limited adolescent period consistently reversed genetic defects in mesofrontal innervation, task-coordinated neuronal activity, and memory-guided decision-making at adulthood. Furthermore, adolescent stimulation of dopamine neurons also reversed the same cognitive deficits in *Disc1*+/- mice. Our findings reveal common mesofrontal circuit alterations underlying the cognitive deficits caused by two different genes and demonstrate the feasibility of adolescent neurostimulation to reverse these circuit and behavioral deficits. These results may suggest developmental windows and circuit targets for treating cognitive deficits in neurodevelopmental disorders.

## eLife assessment

This is an **important** study that addresses the interesting question of whether stimulation of DA input to prefrontal cortex during adolescence can be used to rescue genetic defects on DA regulation of PFC function. The conclusions are **convincingly** supported by the data together with discussion of some limitations of the approach. This story will of interest to a broad group of neuroscientists interested in regulation of prefrontal cortical function in behavior.

## Introduction

Dysfunctions of the dopamine system are commonly implicated in neuropsychiatric disorders such as schizophrenia and substance-use disorders (*Insel, 2010*; *Saha et al., 2005*; *Volkow et al., 2018*). In addition, the treatment effect of antipsychotic medication is related to its effectiveness in blocking

***For correspondence:**
kuanhong_wang@urmc.
rochester.edu

[†]These authors contributed equally to this work

**Competing interest:** The authors declare that no competing interests exist.

dopamine D2 receptors (*Howes and Kapur, 2009*; *Weinstein et al., 2017*). However, psychiatric symptoms include not only psychosis that can be treated by antipsychotics, but also cognitive disabilities in executive functions and memory (*Green, 2007*; *Miyamoto et al., 2005*; *Weinstein et al., 2017*). While current antipsychotic therapies based on D2 antagonism have little effect on cognitive disabilities (*Barbui et al., 2009*; *Davidson et al., 2009*; *Keefe et al., 2004*; *Miyamoto et al., 2005*), frontal cortical dopamine deficits are reported in schizophrenia and the role of frontal cortical dopamine D1 receptor activation in cognitive function has been demonstrated in animal studies (*Arnsten, 1997*; *Castner et al., 2000*; *Slifstein et al., 2015*; *Vijayraghavan et al., 2007*). Unfortunately, due to safety and bioavailability issues, very few D1 receptor agonists have progressed to clinical testing (*Girgis et al., 2016*; *Marder, 2006*; *Tamminga, 2006*). The long-standing difficulty of developing effective pharmacological treatments for cognitive symptoms suggests the necessity of new investigative approaches, such as evaluating the underlying neural circuitry and its potential for modification.

Dopamine regulates a multitude of behavioral functions through different anatomical pathways from dopaminergic neurons in the midbrain ventral tegmental area (VTA) and substantia nigra to cortical and subcortical target regions (*Beier et al., 2015*; *Björklund and Dunnett, 2007*; *Lerner et al., 2015*; *Watabe-Uchida et al., 2012*). The mesofrontal pathway innervates the frontal cortex and an optimal level of dopamine is important for normal frontal circuit functions (*Arnsten et al., 1994*; *Floresco, 2013*; *Robbins, 2000*; *Vander Weele et al., 2018*). The frontal cortical circuit integrates dopaminergic signals with multisensory and memory information to control behavioral actions (*Dalley et al., 2004*; *Fuster, 2008*; *Miller and Cohen, 2001*; *Rushworth et al., 2012*). Together, the frontal cortex and its dopaminergic input form a critical substrate for cognitive functions and a hypoactive frontal cortical dopamine system is found in schizophrenia patients (*Slifstein et al., 2015*; *Weinstein et al., 2017*). Genetic disruptions of several genes involved in synaptic functions related to psychiatric disorders, such as *Arc* and *Disc1*, lead to hypoactive mesofrontal dopaminergic input in mice (*Fromer et al., 2014*; *Managò et al., 2016*; *Niwa et al., 2013*; *Niwa et al., 2010*; *Purcell et al., 2014*; *Wen et al., 2014*). Although there are many differences between these mouse lines and specific human disease states, these mice offer opportunities to test whether genetic deficits in frontal cortex function can be reversed through circuit interventions.

Adolescence is an important period for the development of cognitive control functions and exploration behaviors (*Dahl et al., 2018*; *Larsen and Luna, 2018*; *Spear, 2011*). Cognitive control deficits often emerge during adolescence as a characteristic manifestation of several psychiatric disorders including schizophrenia and substance-use disorders (*Insel, 2010*; *Paus et al., 2008*; *Volkow et al., 2018*). Developmental studies of the mesofrontal dopamine projection in both non-human primates and rodents suggest that this pathway exhibits a protracted maturation through adolescence (*Kalsbeek et al., 1988*; *Lambe et al., 2000*; *Naneix et al., 2013*; *Rosenberg and Lewis, 1995*; *Ye et al., 2017*). Importantly, the structure and function of the mesofrontal circuit are malleable to experience; activity-dependent modification during adolescence and adolescent exposure to substances of abuse can produce long-lasting deficits (*Jobson et al., 2019*; *Mastwal et al., 2014*; *Reynolds et al., 2019*; *Walker et al., 2017*). However, it is unknown whether adolescent intervention in this pathway can induce long-lasting circuit changes conducive for the recovery of cognitive function from genetic deficits.

Mouse genetic models of frontal dopamine deficits and the adolescence malleability of this circuit present opportunities to investigate the cellular substrates underlying cognitive deficits and test potential intervention strategies (*Managò et al., 2016*; *Niwa et al., 2013*; *Niwa et al., 2010*). In this study, using single-cell resolution neuronal ensemble imaging in freely behaving animals, we identified a disruption of neuronal coordination in frontal cortex that is associated with cognitive impairment in mouse models. Furthermore, taking advantage of the activity-dependent adolescent plasticity in the mesofrontal circuit, we developed targeted chemogenetic and optogenetic neuromodulation techniques that are able to reverse the neural coordination and cognitive behavioral deficits in adult animals. Our results demonstrate the capability of adolescent frontal dopamine circuit stimulation to achieve long-term reversal of cognitive deficits and suggest potential translational targets and strategies for psychiatric treatments.

# Results

## Characterization of cognitive and mesofrontal innervation deficits in *Arc* mutant mice

We first used an *Arc* mutant model of a hypofunctioning mesofrontal circuit (*Managò et al., 2016*) for behavioral, anatomical tracing, and neural activity imaging studies. To assess whether genetic disruption of *Arc* (by knocking in a GFP reporter cassette; *Wang et al., 2006*) affects cognitive control of behavior, we examined wild-type and *Arc* homozygous mutant mice in a Y-maze spatial navigation task. This task takes advantage of the innate spatial navigation ability of mice and does not require extensive pretraining, making it well suited for both developmental and adult testing (*Belforte et al., 2010*; *Lalonde, 2002*; *Richman et al., 1986*). Wild-type mice explore the three arms of the maze using

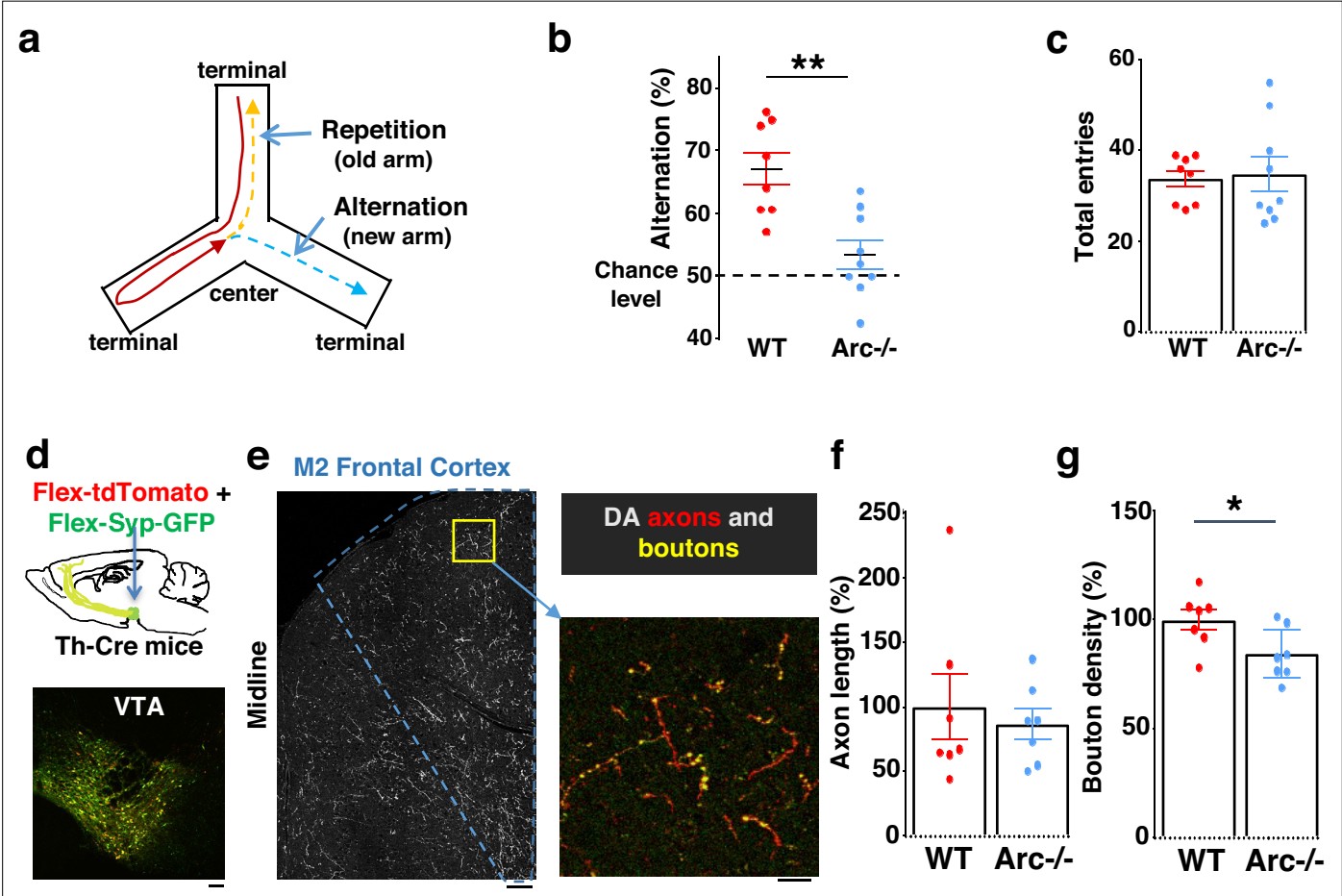

**Figure 1.** Characterization of cognitive and mesofrontal deficits in *Arc* mutant mice. (**a**) Diagram showing the navigation choices for mice in a Y-maze. At the center of the maze, a mouse has a choice to enter either a new arm (alternation) or a previously visited old arm. (**b**) Alternation percentage in the Y-maze task showing significant reduction in the *Arc-/-* animals compared to wild-type animals (**p=0.001, *t*-test, t(15)=3.975, *WT* N=8, *Arc-/-* N=9 mice, both groups passed Shapiro-Wilk normality test at alpha = 0.05). (**c**) Total arm entries are comparable between *Arc-/-* and *WT*. (**d**) Top, schematic for AAV injection in Th-Cre animals to label dopaminergic neurons. Bottom, confocal image showing tdTomato (red) and SypGFP (green) labeling in the ventral tegmental area (VTA). Scale bar, 100 μm. (**e**) Left, confocal image showing labeled dopaminergic axons in the frontal cortex. The dotted line indicates the region-of-interest for M2. Scale bar, 100 μm. Right, zoomed-in region showing labeled axons (tdTomato, red) and boutons (tdTomato+SypGFP, yellow). Scale bar, 20 μm. (**f, g**) The normalized axon length (**f**) is not significantly different. The normalized bouton density (**g**) is significantly reduced in *Arc-/-* animals compared to *WT* (*p=0.034, *t*-test, t(12)=2.393, N=7 mice for each group). The axon length is normalized by the number of labeled cells in VTA, the bouton density is normalized by the axon length, and both are expressed as a percentage of the group average in *WT* mice. All the error bars indicate SEM.

The online version of this article includes the following source data and figure supplement(s) for figure 1:

**Source data 1.** Characterization of cognitive and mesofrontal deficits in *Arc* mutant mice.

**Figure supplement 1.** Chemogenetic inhibition of M2 frontal cortex reduces alternation in Y-maze.

a memory-based decision-making strategy, preferring to visit a new arm instead of the most recently visited arm (*Figure 1a*). The percentage of new arm visits out of the total arm visits is referred to as 'alternation percentage', which is approximately 67% in wild-type mice and significantly above the chance level from random exploration (50%). By contrast, the alternation percentage in *Arc* mutant mice is significantly reduced toward the chance level (p=0.001, *t*-test, t(15) = 3.975, *WT*: 67.2 ± 2.6%, vs. *Arc-/-*: 53.4 ± 2.3%, N=8 and 9 mice, respectively; *Figure 1b*). *Arc* mutant mice showed a similar number of total arm entries as the wild-type mice (*Figure 1c*), suggesting that the *Arc* mutation did not affect the overall locomotor activity or motivation level. Taken together, these results suggest that *Arc* mutant mice have a deficit in memory-guided choice behaviors.

Optimal performance in the Y-maze requires an intact frontal cortex and VTA (*Delatour and Gisquet-Verrier, 1996*; *Divac et al., 1975*; *Pioli et al., 2008*). The M2 region of the frontal cortex plays an important role in action planning, generating the earliest neural signals among frontal cortical regions that are related to upcoming choice during spatial navigation (*Sul et al., 2011*; *Sul et al., 2010*). To verify the involvement of the M2 cortex in Y-maze alternation, we expressed the chemogenetic inhibitor DREADD-Gi (*Roth, 2016*) in M2 with an adeno-associated viral (AAV) vector and used clozapine *N*-oxide (CNO) injection to inhibit M2 neural activity in wild-type mice (*Figure 1—figure supplement 1a*). We found that the alternation percentage was significantly reduced in the DREADD-Gi animals compared with control animals (p=0.017, *t*-test, t(10)=2.845, Ctrl: 64.6 ± 1.8%, Gi: 54.1 ± 3.2%, N=6 for each group, *Figure 1—figure supplement 1b*), while other aspects of motor behaviors in this task were not affected (*Figure 1—figure supplement 1c and d*). These results confirm the involvement of M2 frontal cortical neurons in memory-guided decision-making during the Y-maze task.

Dopaminergic input from the VTA is critical to optimal frontal cortical function in controlling cognitive processes (*Arnsten et al., 1994*; *Floresco, 2013*; *Robbins, 2000*). While the prelimbic area has the highest level of dopaminergic terminals among frontal cortical regions, a robust presence of midbrain dopaminergic projections and dopamine release in the M2 frontal cortex have been well established by immunostaining, viral labeling, single-cell axon-tracing, and in vivo imaging of dopamine biosensors (*Aransay et al., 2015*; *Berger et al., 1991*; *Mastwal et al., 2014*; *Patriarchi et al., 2018*). We therefore examined the anatomical structure of the M2 dopaminergic projection to determine if this input was altered in *Arc* mutant mice.

Mesocortical dopamine neurons have much stronger tyrosine hydroxylase (TH) expression than dopamine transporter expression, which is different from mesostriatal dopamine neurons (*Lammel et al., 2008*; *Lammel et al., 2015*; *Li et al., 2013*; *Sesack et al., 1998*). In addition, TH immunoreactivity in the frontal cortex can label dopaminergic axons originated from the VTA, and ablation of VTA dopaminergic neurons removes this labeling (*Niwa et al., 2013*; *Ye et al., 2017*). Accordingly, Th-Cre transgenic lines have been frequently used to label these neurons and study the mesocortical pathway (*Ellwood et al., 2017*; *Gunaydin et al., 2014*; *Lammel et al., 2012*; *Lohani et al., 2019*; *Vander Weele et al., 2018*).

We injected Cre-dependent AAV vectors into the VTA of Th-Cre mice to label dopaminergic axons with a cytoplasmic marker tdTomato, and the dopamine release sites in axonal boutons with a GFP reporter fused to synaptic vesicle protein synaptophysin (*Figure 1d and e*; *Manion et al., 2022*; *Mastwal et al., 2014*; *Oh et al., 2014*; *Wang et al., 2017*). Although the dopaminergic axon length in the M2 frontal cortex (normalized by the number of labeled cells in VTA) was not significantly different between *Arc* mutant and wild-type animals (*Figure 1f*), we found a significant reduction in the dopaminergic bouton density in this region in *Arc* mutant mice compared to wild-type animals (p=0.034, *t*-test, t(12)=2.393, *WT*: 100 ± 5%, *Arc -/-*: 84 ± 5%, N=7 for each group; bouton density normalized by axon length; *Figure 1g*). These results suggest that dopaminergic innervations of the M2 frontal cortex are reduced in *Arc* mutant mice, which agrees with our previous findings of reduced frontal dopamine release and mesofrontal activity in *Arc-/-* mice (*Managò et al., 2016*). Our new results provide further anatomical evidence for a hypofunctional mesofrontal dopamine circuit in these mice.

## Task-coordinated frontal neuronal ensemble activity is disrupted in *Arc* mutant mice

To determine how M2 neuronal activities during Y-maze performance might be affected by *Arc* mutation and reduced dopaminergic input, we expressed a genetically encoded calcium indicator GCaMP6 (*Chen et al., 2013*) in superficial layer (L2/3) M2 neurons and used a head-mounted miniaturized

microscope (*Ghosh et al., 2011*; *Li et al., 2017*; *Liu et al., 2018*; *Wang et al., 2017*) to image task-related neuronal ensemble activity in adult wild-type and *Arc-/-* mice (*Figure 2a*). The microscope lens was placed above the pial surface rather than inserted into the cortex to avoid damage of M2 neurons, and neuronal activities in superficial cortical layers were imaged (*Figure 2—figure supplement 1a–c*, *Figure 2—video 1*). We found that the activity of individual M2 neurons occurred at various positions along the track during Y-maze navigation (*Figure 2b*). An increased proportion of neurons showed peak activation when the wild-type animal was near the center of the maze before making an arm entry (*Figure 2c*). In contrast, this proportion was significantly reduced in *Arc* mutant mice compared to wild-type animals (p<0.0001, chi-square test, *WT* 938 cells from 8 mice, *Arc-/-* 1338 cells from 7 mice). This effect is not due to a general reduction of neural activity in *Arc-/-* mice, because there is no difference between wild-type and *Arc-/-* mice in the average activity of neurons throughout the task period (*Figure 2—figure supplement 1d–f*). In addition, there is no correlation between the average activity of neurons with alteration behaviors in Y-maze (*Figure 2—figure supplement 1j*). Thus, these results suggest that the coordinated activation of M2 neurons before making a choice is disrupted in *Arc* mutant mice.

To further examine how exploratory choices may be encoded in the activities of individual M2 neurons, we compared the neuronal activity patterns along the navigation trajectories leading to new arm visits versus old arm visits. Neurons selective for alternation were identified by differential neural activation between alternating and non-alternating paths (*Figure 2d*). This analysis showed that the proportion of alternation-selective neurons increased in the wild-type mice as the animal approached the maze center, but this increase was blunted in the *Arc* mutant animals (p<0.0001, chi-square test, *Figure 2e*). These same mutant animals also showed a specific reduction of alternation choices in the Y-maze (p=0.047, *t*-test, t(13)=2.196, *WT*: 67.8 ± 4.4%, N=8 mice; *Arc-/-*: 54.5 ± 4.1%, N=7 mice, *Figure 2—figure supplement 1g–i*), replicating the deficit we observed in separate groups of animals that did not carry the miniaturized microscope (*Figure 1b*). Together, these results demonstrate that both task-related frontal cortical activity and memory-guided decision behavior are impaired in *Arc* mutant mice.

## Adolescent dopamine neuron stimulation leads to long-term reversal of mesofrontal circuit deficits

Given the hypofunctioning mesofrontal dopaminergic circuit in *Arc* mutant mice, we next investigated the possibility of developing neurostimulation strategies to restore normal circuit functions. The dopaminergic innervation in the mesofrontal circuit exhibits a protracted maturation from postnatal day 21 (P21) to P56 (*Hoops and Flores, 2017*; *Kalsbeek et al., 1988*; *Naneix et al., 2012*; *Niwa et al., 2010*). P35–42 is in the center of this period and captures the mid-adolescent stage in rodents (*Spear, 2000*). We have previously shown that increasing dopamine neuron activity by wheel running or optogenetic stimulation during this period, but not adulthood, can induce formation of mesofrontal dopaminergic boutons and enhance mesofrontal circuit activity in wild-type mice (*Mastwal et al., 2014*). We therefore chose the P35–P42 adolescent window to stimulate the mesofrontal dopamine circuit and test the long-term effect of this intervention on the frontal circuit and memory-guided decision-making deficits in adult *Arc* mutant mice.

To stimulate dopamine neuron activity, we expressed chemogenetic activator DREADD-Gq (*Alexander et al., 2009*) in the VTA dopamine cells using a combination of stereotaxically injected Cre-dependent AAV vectors and a Th-Cre transgenic mouse line (*Mastwal et al., 2014*; *Figure 3a*, *Figure 3—figure supplement 1a*). CNO (1 mg/kg) was injected systemically to enhance the activity of DREADD-Gq-expressing dopamine neurons. To control for any potential off-target effect of CNO, another group of Th-Cre mice that expressed AAV-mCherry reporter in dopamine neurons also received CNO injection. Previous studies have shown that DREADD-Gq-induced neural activation reaches its peak approximately 1 hr after CNO injection and returns to baseline approximately 9 hr after injection (*Alexander et al., 2009*). Increased VTA dopamine activity is known to enhance frontal cortical activity, which can be measured with calcium indicator GCaMP6 (*Lavin et al., 2005*; *Managò et al., 2016*; *Mastwal et al., 2014*). We validated the DREADD-Gq-induced enhancement of mesofrontal activity by examining M2 cortical activity with GCaMP6 before and after CNO injection in these mice (*Figure 3a and b*, *Figure 3—figure supplement 1b*). There was a significant increase in the M2 cortical activity measured 1 hr after the injection of CNO in DREADD-Gq animals, whereas

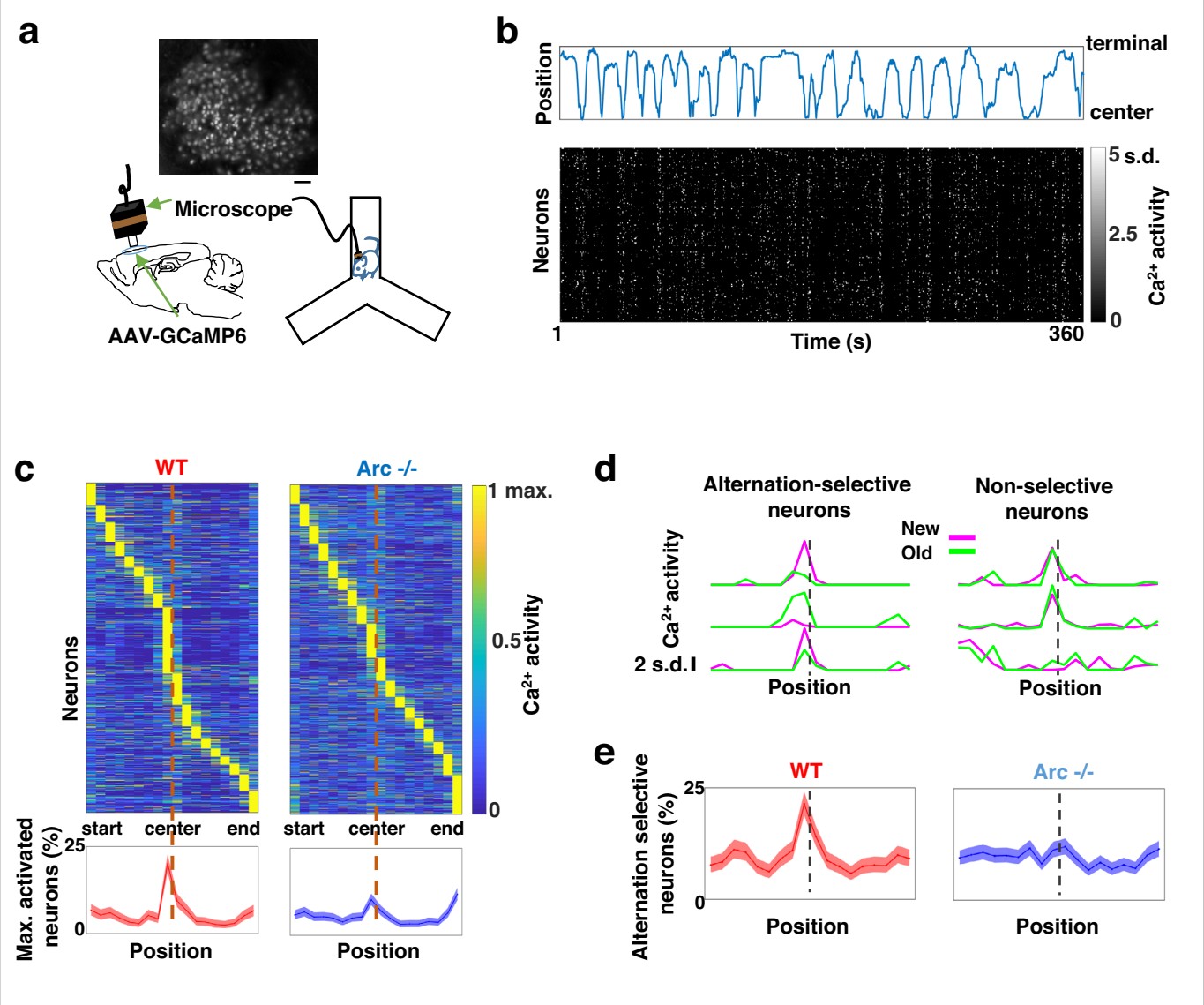

**Figure 2.** Task-coordinated frontal neuronal ensemble activity is disrupted in *Arc* mutant mice. (**a**) Diagram showing the setup for miniaturized microscope imaging of frontal cortical activity in mice performing the Y-maze task. The example image represents the projection (by standard deviation of ΔF/F) of a calcium activity movie (~500 s), showing labeled M2 frontal cortical neurons. Scale bar, 100 μm. (**b**) Top, an example plot showing the positions of a wild-type mouse relative to the Y-maze center during navigation. Bottom, raster plot showing the calcium activity of M2 neurons simultaneously recorded during navigation. (**c**) Top, the average activity of individual frontal cortical neurons at binned positions relative to the center of Y-maze in *WT* (938 neurons from 8 animals) and *Arc-/-* (1338 neurons from 7 animals) animals. Bottom, the proportion of neurons showing maximal activation at each maze position. This proportion peaked right before the maze center in *WT* mice but reduced significantly in *Arc-/-* mice. Shaded areas indicate 95% confidence intervals. (**d**) Examples of neurons showing differential (alternation-selective, traces from three neurons on the left) or similar (non-selective, traces from three neurons on the right) activity between new and old arm visits. (**e**) The proportion of alternation-selective neurons peaked right before the maze center in *WT* mice but reduced significantly in *Arc-/-* mice. Shaded areas indicate 95% confidence intervals.

The online version of this article includes the following video, source data, and figure supplement(s) for figure 2:

**Source data 1.** Task-coordinated frontal neuronal ensemble activity is disrupted in *Arc* mutant mice.

**Figure supplement 1.** Imaging cortical neuronal activity in Y-maze alternation task with a miniaturized head-attached microscope.

**Figure 2—video 1.** Frontal cortical neuronal ensemble activity in freely moving mice during Y-maze exploration.

https://elifesciences.org/articles/87414/figures#fig2video1

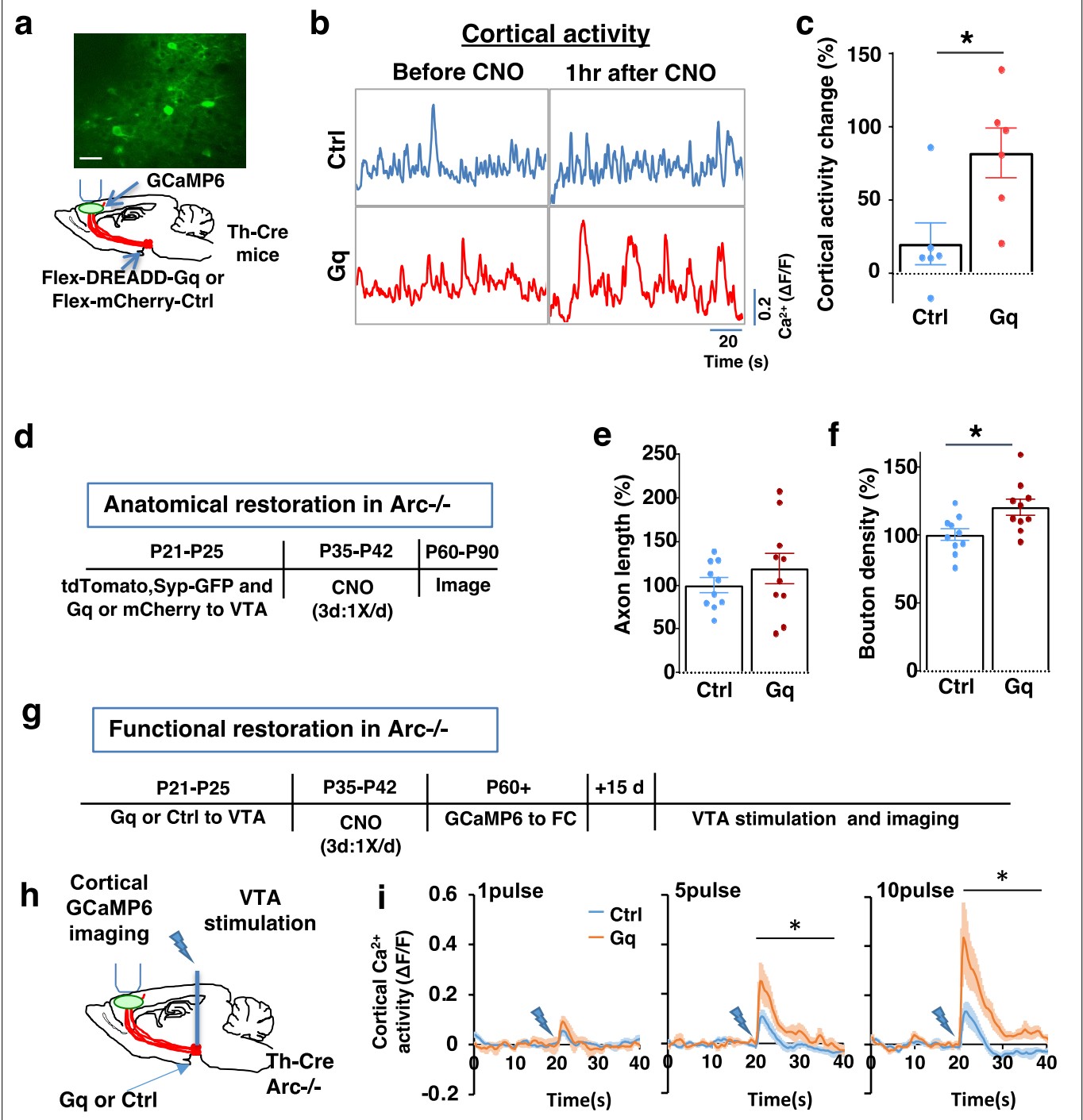

**Figure 3.** Adolescent dopamine neuron stimulation leads to long-term reversal of mesofrontal circuit deficits. (**a**) Schematic for labeling dopamine neurons with DREADD-Gq-mCherry and imaging frontal cortical activity with GCaMP6. The example image at the top shows a two-photon image of labeled cortical neurons. Scale bar, 20 μm. (**b**) Example traces of spontaneous cortical activity averaged from the whole image frame in control mCherry only (blue) and DREADD-Gq (red) animals before and 1 hr after clozapine *N*-oxide (CNO) injection. (**c**) Gq animals show significantly higher change in cortical activity after CNO injection compared to control mCherry only animals, suggesting CNO-induced activation of the mesofrontal circuit (*p=0.018, *t*-test, t(10)=2.814, N=6 mice for each group). Cortical activity is summarized by the standard deviation (SD) of the spontaneous activity traces. Activity change is calculated as (SD2-SD1)/SD1, where SD1 is before and SD2 after treatment. (**d**) Diagram showing experimental procedures to evaluate the effect of adolescent dopamine neuron stimulation on the structure of frontal dopaminergic projections. (**e, f**) Normalized axon length (**e**) is not significantly different. Normalized bouton density (**f**) is significantly increased in Gq animals compared to Ctrl (*p=0.013, *t*-test, t(18)=2.763, N=10 for each group). The axon length is normalized by the number of labeled cells in ventral tegmental area (VTA), the bouton density is normalized by the

*Figure 3 continued on next page*

*Figure 3 continued*

axon length, and both are expressed as a percentage of the group average in Ctrl mice. (**g**) Diagram showing procedures to determine the effect of adolescent dopamine neuron stimulation on mesofrontal circuit activity in *Arc-/-*;Th-Cre mice labeled with DREADD-Gq or mCherry-Ctrl. (**h**) Schematic showing the experimental setup to measure the mesofrontal circuit activity by VTA electrical stimulation and GCaMP6 imaging in the frontal cortex. (**i**) Time courses of cortical calcium signals in response to VTA stimulation in *Arc-/-*;mCherry-Ctrl and *Arc-/-*; DREADD-Gq mice. 1, 5, or 10 pulses of electrical stimuli (50 Hz) were delivered at 20 s after the start of imaging. The cortical calcium activity at each time point is represented by the change in image fluorescence relative to the baseline image fluorescence (ΔF/F). (5 pulse $F_{(1,10)}$=6.0, *p=0.034, 10 pulse $F_{(1,10)}$=9.5, *p=0.012, two-way ANOVA, N=6 mice per group).

The online version of this article includes the following source data and figure supplement(s) for figure 3:

**Source data 1.** Adolescent dopamine neuron stimulation leads to long-term reversal of mesofrontal circuit deficits.

**Figure supplement 1.** Ventral tegmental area (VTA) expression of DREADD-Gq and validation of clozapine *N*-oxide (CNO)-induced cortical activation.

no significant effect occurred in control mCherry animals (DREADD-Gq: 82±17% vs. mCherry Ctrl: 20 ± 14%, p=0.018, *t*-test, t(10)=2.814, N=6 for each group; *Figure 3c*). In addition, saline injection in DREADD-Gq-expressing mice did not alter M2 neural activity, and CNO-induced increase in M2 neural activity was suppressed by D1 antagonist SCH23390 (p=0.0002, N=7 for each group, *Figure 3—figure supplement 1c*). These results confirmed CNO-induced activation of the mesofrontal dopaminergic circuit in DREADD-Gq animals.

After the validation of activity enhancement by chemogenetic stimulation of dopamine neurons, we examined if this treatment would have any long-term effect on the circuit deficits in *Arc* mutant mice. Our previous work in wild-type adolescent mice showed that a single optogenetic stimulation session or a 2 hr wheel running session is sufficient to induce bouton formation in mesofrontal dopaminergic axons (*Mastwal et al., 2014*). In this study, we sought to rescue existing structural and functional deficits in the mesofrontal dopaminergic circuits due to genetic mutations. Because previous studies suggested that an optimal level of dopamine is important for normal cognitive function (*Arnsten et al., 1994*; *Floresco, 2013*; *Robbins, 2000*), we elected to do multiple stimulation sessions to boost the potential rescue effects.

We first tested whether the structural deficits in the mesofrontal dopaminergic circuit would be affected by adolescent dopamine neuron stimulation. DREADD-Gq or mCherry expression was combined with tdTomato and synaptophysin-GFP to label dopaminergic axons and boutons in the *Arc* mutant animals (*Arc-/-*;Th-Cre). After three CNO treatments in adolescence (one injection per day for 3 days in mice 5 weeks of age in their home cages), the mice were kept in their regular home cage until adulthood (8 weeks) for histological assessment (*Figure 3d*). Although the mesofrontal dopamine axon length did not show any difference (*Figure 3e*), we found a significant increase in dopaminergic bouton density in DREADD-Gq mice compared to control animals (Ctrl: 100±4% vs. Gq: 120 ± 6%, p=0.013, *t*-test, t(18)=2.763, N=10 for each group; *Figure 3f*). These results suggest that adolescent dopamine neuron stimulation in *Arc* mutant mice led to a long-term enhancement of the connection between dopaminergic axons and M2 frontal cortical neurons, thereby ameliorating the structural deficits in these mice.

We next tested whether adolescent dopamine neuron stimulation would also lead to long-term improvement of the functional deficits in the mesofrontal circuit of *Arc* mutant mice. Our previous work indicated that *Arc* mutants have reduced frontal cortical response to VTA stimulation (*Managò et al., 2016*). Using the same assay in adult mice, we found that the cortical calcium activity in response to VTA electrical stimulation was significantly enhanced in DREADD-Gq mice compared to control mCherry mice after adolescence CNO treatment (10 pulse $F_{(1,10)}$=9.513, p=0.012, two-way ANOVA, N=6 mice per group, *Figure 3g–i*). These results suggest that adolescent dopamine neuron stimulation also leads to long-term improvement of the functional deficits in the mesofrontal dopaminergic circuit of *Arc* mutants.

## Adolescent dopamine neuron stimulation leads to restoration of coordinated frontal neuronal activity and cognition in adulthood

To further examine whether the uncoordinated frontal neural activity patterns in *Arc* mutant mice during the Y-maze task would be renormalized by this adolescent dopamine neuron stimulation strategy, we conducted calcium imaging using the head-mounted miniature microscope in these mice (*Figure 4a*). We found that the coordinated activation of neurons near the maze center

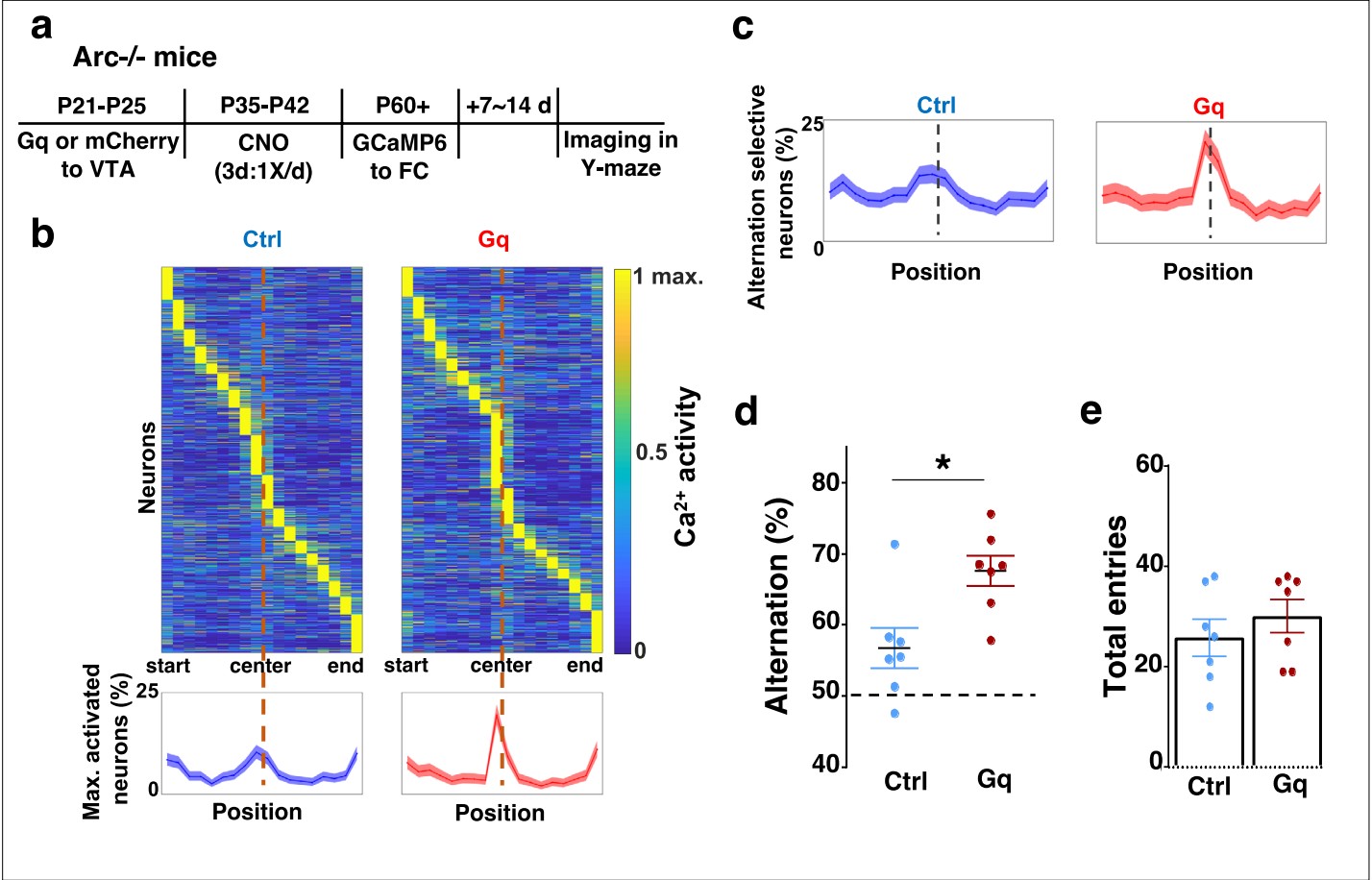

**Figure 4.** Adolescent dopamine neuron stimulation leads to restoration of coordinated frontal neuronal activity and cognition in adulthood. (**a**) Diagram showing procedures for adolescent stimulation of dopamine neurons and functional imaging of frontal cortical neuron activity in adult *Arc-/-*;Th-Cre mice. (**b**) The average activity of individual frontal cortical neurons at binned positions relative to the center of Y-maze in *Arc -/-*; mCherry-Ctrl (1288 neurons from 7 mice) and *Arc-/-*; DREADD-Gq (1008 neurons from 7 mice) animals. Bottom, the proportion of neurons showing maximal activation at each maze position. This proportion reached a higher peak right before the maze center in *Arc-/-*; DREADD-Gq mice compared to *Arc-/-*; mCherry-Ctrl mice. Shaded areas indicate 95% confidence intervals. (**c**) The proportion of alternation-selective neurons reached a significantly higher peak right before the maze center in *Arc-/-*; DREADD-Gq compared to *Arc-/-*; mCherry-Ctrl mice. Shaded areas indicate 95% confidence intervals. (**d**) Y-maze alternation percentage for the animals used in the miniaturized microscope imaging experiments shows significant increase in *Arc-/-*;DREADD-Gq animals compared to *Arc-/-*; mCherry-Ctrl (*p=0.010, *t*-test, t(12)=3.043, N=7 mice for each group, both groups passed Shapiro-Wilk normality test at alpha = 0.05). (**e**) Total arm entries are not significantly different (p=0.399, *t*-test, t(12)=0.875). All the error bars indicate SEM.

The online version of this article includes the following source data and figure supplement(s) for figure 4:

**Source data 1.** Adolescent dopamine neuron stimulation leads to restoration of coordinated frontal neuronal activity and cognition in adulthood.

**Figure supplement 1.** Average activity of M2 neurons during Y-maze exploration.

(*Figure 4b*) and the proportion of alternation-selective neurons (*Figure 4c*) were both significantly enhanced in *Arc* mutant mice that received the DREADD-Gq activation compared to the mCherry control group (p<0.0001, chi-square tests, DREADD-Gq 1008 cells from 7 mice, mCherry 1286 cells from 7 mice). In contrast, the average activity of M2 neurons throughout the task period was not affected by adolescent DREADD-Gq stimulation (*Figure 4—figure supplement 1a–c*). Strikingly, the DREADD-Gq-stimulated animals also showed an enhancement of behavioral alternation (p=0.010, *t*-test, t(12)=3.043, Ctrl: 56.8 ± 2.8%, Gq: 67.6 ± 2.2%, N=7 each group, *Figure 4d*), reaching a level comparable to that in the wild-type mice (*Figure 1b*). Total arm entries were not affected (*Figure 4e*), suggesting no alternations in general locomotor activity. These results indicate that adult task-related functional activity in the M2 frontal cortex is restored by adolescent dopamine neuron stimulation in *Arc* mutant mice.

## Efficacy requirements for adolescent dopamine neuron stimulation

To further characterize the experimental conditions that are important for the restoration of memory-guided decision-making behavior in *Arc* mutant mice, we examined several variables including post-stimulation test interval, stimulation duration, and the age of stimulation. First, we replicated the effects of adolescent dopamine neuron stimulation on Y-maze navigation in adulthood in another cohort of *Arc* mutant mice (p=0.023, *t*-test, t(12)=2.598, Ctrl: 59.5 ± 2.4%, Gq: 68.5 ± 2.5%, N=7 for each group) (*Figure 5a*). Furthermore, when tested only 1 day after the 3-day CNO treatment procedure (one injection per day) in adolescence, *Arc* mutant mice already showed an increased alternation percentage (p=0.031, *t*-test, t(20)=2.322, Ctrl: 55.8 ± 2.9%, Gq: 64.7 ± 2.5%, N=11 for each group) (*Figure 5b*), which is comparable to the effect observed 1 month after treatment in adult mice (*Figure 5a*). These results suggest that the adolescent neurostimulation effect on cognitive behavioral improvement is both fast acting and long lasting.

Second, we also tested a more intense CNO stimulation procedure involving two injections per day for 3 weeks (5 days per week) starting in adolescence. However, this strong 3-week stimulation did not enhance behavioral alternation and appeared to decrease it even further (*Figure 5c*). These results indicate that moderate but not excessive stimulation of dopamine neurons can provide functional improvement of a deficient mesofrontal circuit, consistent with previous studies showing that an optimal level of dopamine is important for normal cognitive function (*Arnsten et al., 1994*; *Floresco, 2013*; *Robbins, 2000*).

Third, we found that in adult *Arc* mutant mice with DREADD-Gq expression, 3-day CNO injection did not lead to any effects on Y-maze alternation behavior assayed 1 month later (*Figure 5d*). These results indicate that adolescent, but not adult, intervention is critical for reversing this behavioral dysfunction, in agreement with the elevated adolescent structural and functional plasticity reported for the mesofrontal circuit (*Mastwal et al., 2014*). In addition, because 3 weeks of adolescent CNO treatment or 3 days of adult CNO treatment in DREADD-Gq mice did not lead to any rescue effects, DREADD-Gq expression alone is unlikely to generate any behavior improvement. Taken together, our findings suggest that a brief stimulation of midbrain dopamine neurons in adolescence has a long-lasting effect to reverse the memory-guided decision-making deficits in *Arc* mutant mice, while not affecting other aspects of motor behaviors in the Y-maze (*Figure 5—figure supplement 1a–d*).

## Specific stimulation of adolescent frontal dopaminergic axons reverses both cognitive and psychomotor deficits

To further evaluate the impact of adolescent dopamine neuron stimulation on another frontal cortex-dependent behavioral deficit in *Arc* mutant mice, we subjected these mice to an amphetamine-induced hyperactivity test, which is considered an animal model for drug-induced psychomotor symptoms (*Forrest et al., 2014*; *Managò et al., 2016*). Classical neuropharmacological studies have shown that amphetamine-induced locomotor activity is facilitated by dopaminergic input to the nucleus accumbens but inhibited by dopaminergic input to the frontal cortex (*Chambers et al., 2003*; *Le Moal and Simon, 1991*; *Tzschentke, 2001*). Consistent with a hypoactive frontal dopamine input, *Arc* mutant mice showed hyper-reactivity to amphetamine compared to the wild-type mice (*Figure 6—figure supplement 1a*) as reported before (*Managò et al., 2016*).

However, we found that adolescent stimulation of VTA dopamine neurons through DREADD-Gq did not reduce the hyper-reactivity to amphetamine in adult *Arc* mutant mice (*Figure 6—figure supplement 1b*). Dopamine neurons in the VTA include separate neuronal populations that project to the frontal cortex and the nucleus accumbens, respectively (*Björklund and Dunnett, 2007*; *Lammel et al., 2008*). Our results suggest that adolescent stimulation of both populations of VTA neurons in mesofrontal and mesoaccumbens pathways, which have opposing effects on amphetamine-induced locomotor activity, result in no net change in this behavioral phenotype at adulthood.

Subsequently, we tested whether targeted stimulation of dopaminergic axons projecting to the frontal cortex would be a better strategy to reinstate memory-guided decision-making in the Y-maze task and prevent amphetamine hyper-reactivity in *Arc* mutant mice. Although CNO in principle could be injected directly into the frontal cortex to stimulate DREADD-Gq-labeled dopaminergic axons, we found in preliminary experiments that repeated injections (three times) in adolescent brain led to tissue deformation and damage. Therefore, we turned to a less invasive optogenetic method to

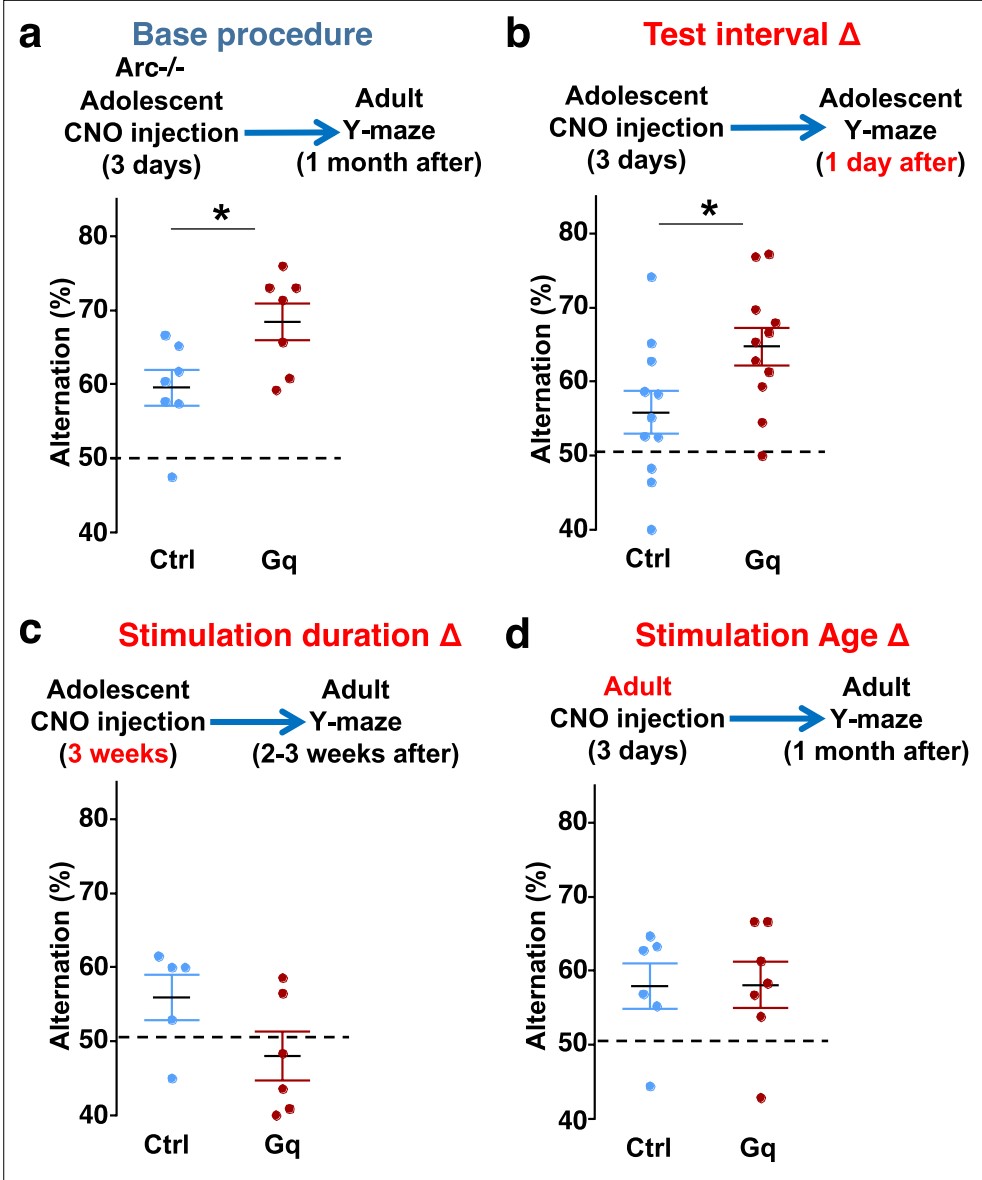

**Figure 5.** Efficacy requirements for adolescent dopamine neuron stimulation. (**a**) Top, diagram showing the base procedure for the stimulation of midbrain dopamine neurons and Y-maze testing in *Arc-/-*;Th-Cre mice labeled with DREADD-Gq or mCherry control viruses. Animals were injected with clozapine *N*-oxide (CNO) (1 mg/kg) once per day for 3 days in adolescence (5 weeks of age) and then tested in the Y-maze at adulthood (~1 month later). *Arc-/-*; DREADD-Gq animals show significantly higher alternation compared to *Arc-/-*; mCherry-Ctrl at adulthood (*p=0.023, *t*-test, t(12)=2.598, N=7 mice for each group, both groups passed Shapiro-Wilk normality test at alpha = 0.05). (**b**) The behavioral effect at the test interval of 1 day after CNO injection. *Arc-/-*; DREADD-Gq animals show significantly higher alternation compared to *Arc-/-*; mCherry-Ctrl (*p=0.031, *t*-test, t(20)=2.322, N=11 mice for each group, both groups passed Shapiro-Wilk normality test at alpha = 0.05). (**c**) The behavioral outcome following a long duration stimulation starting in adolescence (2 times per day, 5 days per week, for 3 weeks). These animals did not show any improvement but a declining trend in performance (p=0.12, *t*-test, t(9)=1.727, *Arc-/-*; mCherry-Ctrl N=5, *Arc-/-*; DREADD-Gq, N=6 mice, both groups passed Shapiro-Wilk normality test at alpha = 0.05). (**d**) The behavioral outcome following 3 days of CNO stimulation administered in adult mice (2–3 months). These animals did not show significant difference (p=0.97, *t*-test, t(11)=0.037, *Arc-/-*;mCherry-Ctrl, N=6; *Arc-/-*;DREADD-Gq, N=7 mice, both groups passed Shapiro-Wilk normality test at alpha = 0.05).

The online version of this article includes the following source data and figure supplement(s) for figure 5:

**Source data 1.** Efficacy requirements for adolescent dopamine neuron stimulation.

**Figure supplement 1.** Behavioral characterization of *Arc-/-* mice under different chemogenetic stimulation conditions.

enhance the activity of labeled axons using stabilized step function opsins (SSFOs), which can increase neural excitability over a 30 min time scale in response to a 2 s pulse of light (*Yizhar et al., 2011*).

To validate the effect of SSFO-based stimulation, we expressed Cre-dependent SSFO in the midbrain dopamine neurons and stimulated their axonal projections in the M2 frontal cortex with blue light through a cranial window (*Figure 6a*; *Figure 6—figure supplement 1c–d*). We monitored M2 neural activity using calcium reporter GCaMP6 before and 30 min after light activation. Similar to the DREADD-Gq-mediated activity changes in the mesofrontal circuit, frontal cortical activity was increased after light activation compared to before in SSFO animals, and this increase was significantly higher than the control EGFP animals that were also exposed to the light (p=0.001, *t*-test, t(8)=5.135, Ctrl: 7.5 ± 3.1%, SSFO: 46.7 ± 6.9%, N=5 each group, *Figure 6b–c*, *Figure 6—figure supplement 1e*). These results indicate that light stimulation of SSFO-labeled frontal dopamine axons can enhance the activity in the mesofrontal circuit.

To determine whether adolescent stimulation of mesofrontal dopamine axons would be sufficient to reinstate memory-guided decision-making in *Arc* mutant mice, we stimulated SSFO-labeled dopamine axons in the frontal cortex once per day for 3 days in adolescence (*Figure 6d*). When tested either 1 day after light stimulation or 1 month after in adulthood, SSFO-labeled mice showed a significant increase in the Y-maze alternation percentage compared to control EGFP animals (*Figure 6e*, 1 day after, p=0.028, *t*-test, t(15)=2.440, Ctrl: 58.4 ± 1.4%, SSFO: 66.5 ± 3.2%; *Figure 6g*, adulthood, p=0.015, *t*-test, t(15)=2.748, Ctrl: 55.5 ± 2.5%, SSFO: 66.7 ± 3.3%; N=9 for Ctrl, N=8 for SSFO). On the other hand, total arm entries were not affected (*Figure 6f and h*). These results suggest that similar to VTA dopamine neuron stimulation, projection-specific stimulation of dopaminergic axons in the frontal cortex is sufficient to restore memory-guided decision-making in *Arc* mutant animals.

Moreover, SSFO-mediated frontal dopamine axon stimulation in adolescence also significantly reduced the hyper-reactivity to amphetamine in adult *Arc* mutant mice (p=0.037, F(1,14)=5.3, two-way ANOVA, N=8 each, *Figure 6i*). Thus, adolescent stimulation of dopaminergic projections in the frontal cortex provides an effective strategy for reversing frontal dysfunctions in both memory-guided decision-making and psychostimulant reactivity.

## Adolescent dopamine neuron stimulation reverses cognitive deficits in *Disc1* mutant mice

Finally, we sought to determine if this adolescent neurostimulation strategy would be applicable to another genetic model with a hypoactive mesofrontal circuit. Previous studies showed that knocking down *Disc1* or overexpressing a dominant negative *Disc1* reduced mesofrontal dopaminergic innervation, dopamine release, and cognitive dysfunction (*Niwa et al., 2013*; *Niwa et al., 2010*). Using a *Disc1* mouse model that contains a knocked-in mutation (*Disc1+/-*) identified from a subset of human patients (*Kim et al., 2021*; *Wen et al., 2014*), we first examined whether mesofrontal activity is also impaired in this model by imaging the calcium activity of M2 frontal cortex with two-photon microscopy in response to VTA stimulation (*Figure 7a*). Our results showed a reduction of VTA-induced frontal cortical activity in *Disc1* mutants compared to wild-type animals (10 pulse F(1,10)=16.5, p=0.002, two-way ANOVA, N=6 mice per group, *Figure 7b*), indicating a mesofrontal hypofunction in this *Disc1* mutant line consistent with other *Disc1* models (*Niwa et al., 2013*; *Niwa et al., 2010*).

To further determine if our *Disc1* mutants have deficits in memory-guided decision-making, we conducted the Y-maze test and found reduced alternation in the *Disc1* mutants compared to the wild-type animals (p=0.044, *t*-test, t(18)=2.171, *WT*: 66.8 ± 2.6%, *Disc1+/-*: 58.9 ± 2.5%, N=10 for each group; *Figure 7c*). Thus, similar to *Arc* mutant mice, *Disc1* mutant mice also exhibit a hypofunctional mesofrontal circuit and impaired cognitive function.

We then tested whether the same adolescent neurostimulation strategy used in the *Arc* mutant mice would also reverse the cognitive deficits in *Disc1* mutants. Remarkably, we found that the alternation percentage in adult *Disc1* animals was significantly increased after adolescent neurostimulation (p=0.012, *t*-test, t(9)=3.139, Ctrl: 60.7 ± 2.4%, N=5, Gq: 69.3 ± 1.5% N=6, *Figure 7d*), reaching a level similar to the wild-type mice. Other aspects of motor behaviors in the Y-maze were not affected (*Figure 7—figure supplement 1a–b*). Taken together, these results suggest that adolescent dopamine neuron stimulation is effective for long-term cognitive improvement in two different genetic models of mesofrontal hypofunction.

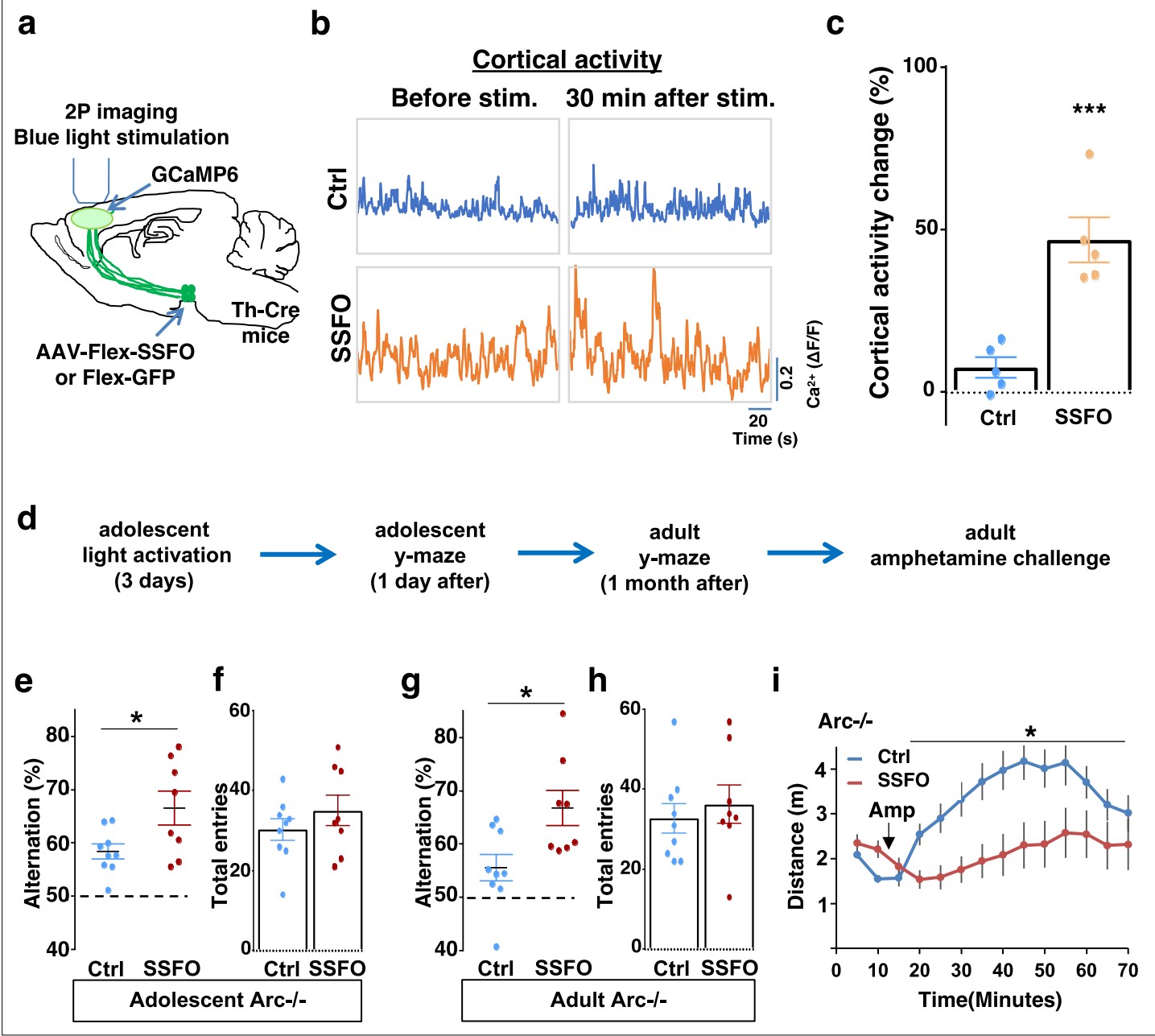

**Figure 6.** Specific stimulation of adolescent frontal dopaminergic axons leads to reversal of both cognitive and psychomotor deficits. (**a**) Schematic for labeling dopamine neurons with stabilized step function opsin (SSFO) and two-photon imaging of frontal cortical activity with GCaMP6. (**b**) Example traces of spontaneous cortical activity averaged from the whole image frame in control EGFP only (blue) and SSFO (orange)-labeled animals before and 30 min after frontal cortical blue light stimulation. (**c**) SSFO animals show significantly higher change in cortical activity (summarized by the standard deviation of spontaneous activity traces) compared to control EGFP animals after the light activation, suggesting light activation of SSFO expressing dopamine neurons (***p=0.001, *t*-test, t(8)=5.135, N=5 mice for each group). Cortical activity is summarized by the standard deviation (SD) of the spontaneous activity traces. Activity change is calculated as (SD2-SD1)/SD1, where SD1 is before and SD2 after treatment. (**d**) Diagram showing procedures for local light activation of frontal dopaminergic projections and Y-maze testing in *Arc-/-*;Th-Cre mice labeled with SSFO or Ctrl-GFP viruses. Light activation was delivered in adolescence (5 weeks of age) once per day for 3 days. Animals were first tested in the Y-maze 1 day after the last light activation and then tested again in the Y-maze at adulthood, followed by an amphetamine-induced locomotion test. (**e–h**) SSFO-expressing animals show significantly higher alternation compared to control animals 1 day after light activation (**e**) (*p=0.028, *t*-test, t(15)=2.440, N=9 EGFP, N=8 SSFO mice, both groups passed Shapiro-Wilk normality test at alpha = 0.05), with no difference in total entries (**f**). These animals also show higher alternation at adulthood (**g**) (*p=0.015, *t*-test, t(15)=2.748, EGFP N=9, SSFO N=8, both groups passed Shapiro-Wilk normality test at alpha = 0.05) with no difference in total entries (**h**). (**i**) In *Arc-/-*;Th-Cre mice that received adolescent frontal light stimulation, amphetamine-induced locomotion is significantly

*Figure 6 continued on next page*

*Figure 6 continued*

reduced at adulthood in SSFO animals compared to GFP control animals (F(1,14)=5.3, *p=0.037, two-way ANOVA, N=8 mice per group). All the error bars indicate SEM.

The online version of this article includes the following source data and figure supplement(s) for figure 6:

**Source data 1.** Specific stimulation of adolescent frontal dopaminergic axons leads to reversal of both cognitive and psychomotor deficits.

**Figure supplement 1.** Amphetamine hyper-reactivity in *Arc-/-* mice and stabilized step function opsin (SSFO) expression and validation.

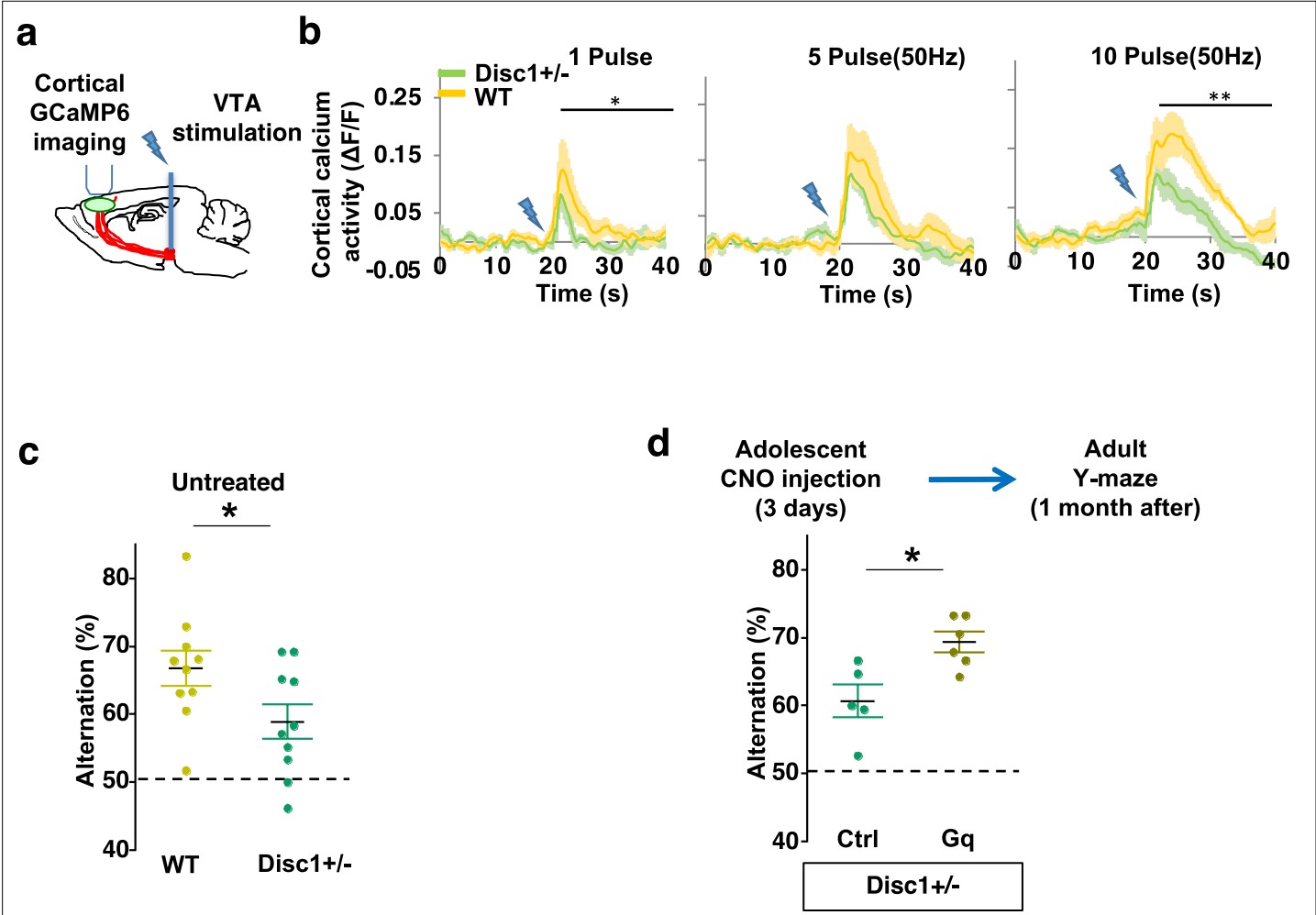

**Figure 7.** Adolescent dopamine neuron stimulation reverses cognitive deficits in *Disc1* mutant mice. (**a**) Schematic showing the experimental setup to measure the mesofrontal circuit activity by ventral tegmental area (VTA) electrical stimulation and GCaMP6 imaging in the frontal cortex. (**b**) Time courses of cortical calcium signals in response to VTA stimulation in *WT and Disc1+/-* mice. 1, 5, or 10 pulses of electrical stimuli (50 Hz) were delivered at 20 s after the start of imaging. The cortical calcium activity at each time point is represented by the change in image fluorescence relative to the baseline image fluorescence (ΔF/F) (1 pulse F(1,10)=5.7, *p=0.038; 10 pulse F(1,10)=16.5, **p=0.002, two-way ANOVA, N=6 per group). (**c**) *Disc1+/-* mice show significant reduction in Y-maze alternation compared to *WT* (*p=0.044, *t*-test, t(18)=2.171, N=10 mice for each group, both groups passed Shapiro-Wilk normality test at alpha = 0.05). (**d**) *Disc1+/-;Th-Cre* mice with DREADD-Gq-labeled dopamine neurons show significantly higher alternation in the Y-maze at adulthood compared to mCherry-Ctrl-labeled mice after 3-day adolescent clozapine *N*-oxide (CNO) injections (*p=0.012, *t*-test, t(9)=3.139, *Disc1+/-*; mCherry N=5, *Disc1+/-;DREADD-Gq*, N=6 mice, both groups passed Shapiro-Wilk normality test at alpha = 0.05). All the error bars indicate SEM.

The online version of this article includes the following source data and figure supplement(s) for figure 7:

**Source data 1.** Adolescent dopamine neuron stimulation reverses cognitive deficits in *Disc1* mutant mice.

**Figure supplement 1.** Characterization of Y-maze behavior in *Disc1+/-* mice.

## Discussion

### Fast-acting and long-lasting cognitive rescue by adolescent dopamine circuit stimulation

In this study, we have demonstrated a powerful role of adolescent dopaminergic input in inducing long-lasting circuit reorganization to reverse genetic deficits in frontal cortex function. Previous studies of dopamine function have focused on its transient neuromodulatory actions on neural dynamics or involvement in reinforcement learning (*Lohani et al., 2019*; *Schultz, 2007*; *Seamans and Yang, 2004*; *Tritsch and Sabatini, 2012*). Here, by examining the effect of dopamine neuron stimulation over a developmental time scale, we have identified both fast acting and persistent effect of adolescent dopamine neuron activity at neuroanatomical, neurophysiological, and behavioral levels. The cognitive dysfunction present in adolescent mutant mice is readily reversed 1 day after neurostimulation and the restored performance level is maintained even 1 month later at adulthood. Interestingly, the behavioral improvement afforded by dopamine neuron stimulation is observed when the intervention was introduced transiently during the adolescence period, but not adulthood, suggesting a sensitive and limited time window for plasticity-inducing therapeutic interventions within mesofrontal dopamine circuits. These results may also stimulate future research to identify the molecular and cellular mechanisms controlling the adolescent plasticity window and explore the possibility and therapeutic potential of reopening plasticity in adulthood (*Caballero and Tseng, 2016*; *Hoops and Flores, 2017*; *Mastwal et al., 2016*; *Mastwal et al., 2014*).

We used both chemogentic (DREADD-Gq) and optogenetic (SSFO) methods to deliver adolescent dopamine circuit stimulation. We did not measure the precise firing patterns of the dopaminergic neurons targeted by SSFO but evaluated the effects of SSFO activation on the frontal cortex. Similar to DREADD-Gq-mediated activity changes in the mesofrontal circuit, which was blocked by D1 antagonist, we found increased frontal cortical activity post-light stimulation of frontal dopamine axons in our SSFO-treated animals. While quantitatively the firing patterns of DREADD-Gq and SSFO-activated dopaminergic neurons likely differ, qualitatively both of these manipulations lead to increased mesofrontal circuit activity and improvements in cognitive behaviors. In our previous work with wild-type adolescent mice, both wheel running and a single 10 min session of phasic optogenetic stimulation of the VTA resulted in dopaminergic bouton outgrowth in the frontal cortex (*Mastwal et al., 2014*). Taken together, these results suggest that adolescent dopaminergic mesofrontal projections are highly responsive to neural activity changes and a variety of adolescent stimulation paradigms are sufficient to elicit lasting changes in this circuit. As dopamine's effects often display an inverted-U dose-response curve (*Floresco, 2013*; *Vijayraghavan et al., 2007*), it will be interesting for future research to compare the effects of specific stimulation methods between wild-type mice and mutant mice with underlying dopamine deficiency. In addition, although we targeted dopamine neurons in our adolescent stimulation, the final behavioral outcome likely includes contributions from co-released neurotransmitters such as glutamate and non-dopaminergic neurons via network effects (*Lohani et al., 2019*; *Morales and Margolis, 2017*), which will be interesting directions for future research.

### Cellular alterations in the mesofrontal circuit underlying cognitive dysfunction and rescue

Our examination across different mechanistic levels suggests a coherent picture in which increased dopamine neuron activity in adolescence leads to enhanced frontal dopaminergic innervation and cortical response to dopamine. Although many of dopamine boutons are not associated with defined postsynaptic structures, these axonal boutons and the active zones they contain are the major release sites for dopamine (*Arbuthnott and Wickens, 2007*; *Goldman-Rakic et al., 1989*; *Liu et al., 2021*; *Sulzer et al., 2016*). Past studies have established a consistent association between increased dopaminergic innervation in the frontal cortex and an increase in dopamine levels (*Naneix et al., 2012*; *Niwa et al., 2010*). Our previous work also found that increasing dopaminergic boutons through adolescent VTA stimulation led to prolonged frontal local field potential responses with high-frequency oscillations (*Mastwal et al., 2014*), which is characteristic of increased dopaminergic signaling (*Gireesh and Plenz, 2008*; *Lewis and O'Donnell, 2000*; *Lohani et al., 2019*; *Wood et al., 2012*). Importantly, in our quantification of the structural changes in this study, we evaluated boutons which were labeled with synaptophysin, a molecular marker indicating the presence of synaptic vesicle release machinery (*Li*

*et al., 2010*; *Oh et al., 2014*). Thus, our study, taken in the context of the previous work, suggests the increased number of boutons signifying an increase in dopaminergic signaling within the mesofrontal circuit.

Previous electrophysiological studies have suggested that dopaminergic signaling can increase the signal-to-noise ratio and temporal synchrony of neural network activity during cognitive tasks (*Lohani et al., 2019*; *Rolls et al., 2008*; *Vander Weele et al., 2018*). The neural activity deficit in *Arc-/-* mice is manifested in the task-coordinated neuronal activation at the decision-making point, but not in the average activity throughout the task. The uncoordinated neural activation pattern in *Arc-/-* mice may contribute to increased noise and decreased signal for decision-making, consistent with the hypodo-paminergic cortical state in *Arc-/-* mice and the computational role of dopamine in enhancing the signal-to-noise ratio of neuronal ensemble activities (*Lohani et al., 2019*; *Mukherjee et al., 2019*; *O'Donnell, 2011*; *Rolls et al., 2008*; *Seamans and Yang, 2004*; *Tseng and O'Donnell, 2007*; *Vander Weele et al., 2018*). After adolescent dopamine neuron stimulation, frontal dopaminergic innervation and cortical response to dopamine are enhanced, which help to restore task-coordinated neuronal activation in adult animals. As coordinated neuronal activity in frontal cortex is implicated in cognitive functions (*Ito et al., 2015*; *Sigurdsson et al., 2010*), the restoration of this activity may underlie the rescue of the cognitive behavioral deficit in *Arc* mutant mice.

It is important to note that dopamine can act on five different receptors expressed in both excit-atory and inhibitory postsynaptic neurons (*O'Donnell, 2010*; *Seamans and Yang, 2004*; *Tseng and O'Donnell, 2007*), and the frontal GABAergic inhibitory network undergoes major functional remod-eling during adolescence (*Caballero and Tseng, 2016*). The developmental increases in dopaminergic innervation to the frontal cortex and local GABAergic transmission are likely synergistic processes, which both contribute to the maturation of high-order cognitive functions supported by the frontal cortex (*Caballero and Tseng, 2016*; *Larsen and Luna, 2018*). Adolescent stimulation of dopamine neurons may interact with this maturational process to promote a network configuration conducive for synchronous and high signal-to-noise neural computation (*Mukherjee et al., 2019*; *Murty et al., 2016*; *Porter et al., 1999*). The microcircuit mechanisms underlying adolescent dopamine stimulation-induced changes, particularly in the GABAergic inhibitory neurons, will be an exciting direction for future research.

## Reversal of cognitive deficits in independent genetic models

Our studies have shown that the mesofrontal dopamine circuit is a common target disrupted by mutations in *Arc* and *Disc1*. The initial motivation of this study was to test if adolescent dopamine stimulation can rescue the deficits in the mesofrontal dopaminergic circuit and cognitive function of *Arc-/-* mice, which were identified in our previous studies (*Managò et al., 2016*). We first conducted multiple levels of analyses including viral tracing, in vivo calcium imaging, and behavioral tests to establish the coherent impacts of adolescent dopamine neuron stimulation on circuits and behaviors. We then examined a range of stimulation protocols to assess the efficacy requirements for cogni-tive improvement, which is our primary goal. Finally, we included *Disc1* mice in our study to test if adolescent dopamine stimulation can also reverse the cognitive deficit in another genetic model for mesofrontal dopamine deficiency. By demonstrating a similar cognitive recuse effect of adolescent VTA stimulation in an independent mouse model, this study provides a foundation for future research to compare the detailed cellular mechanisms that underlie the functional rescue in different genetic models.

Arc and Disc1 have different protein interaction partners and function in distinct molecular path-ways. Arc is best understood for its role in regulating the trafficking of excitatory neurotransmitter receptors at synapses, whereas Disc1 has been shown to act as a scaffold protein to interact with multiple cytoskeletal proteins and synaptic molecules (*El-Boustani et al., 2018*; *Ishizuka et al., 2006*; *Kirov et al., 2012*; *Zhang et al., 2015*). Dopamine-related deficits have been reported in multiple *Arc* and *Disc1* mutant lines or direct knockdown of these molecules (*Gao et al., 2019*; *Managò et al., 2016*; *Niwa et al., 2013*; *Niwa et al., 2010*; *Penrod et al., 2019*; *Salery et al., 2017*), but the exact molecular mechanisms underlying these deficits and the varying severity in different models remain to be elucidated in future studies. Considering the widely reported roles of Arc and Disc1 in regulating activity-dependent synaptic plasticity (*Shepherd and Bear, 2011*; *Tropea et al., 2018*), mutations in these genes may compromise activity-dependent maturation of the mesofrontal dopaminergic circuit.

In the two mouse lines we tested, both *Arc* and *Disc1* mutations lead to a hypoactive mesofrontal circuit and deficits in memory-guided decision-making behaviors. Our study demonstrates that the cognitive behavioral phenotypes arising from distinct genetic origins can be effectively rescued by the same neurostimulation strategy targeting a convergent mesofrontal circuit phenotype during a critical adolescent window.

## Reversal of both cognitive and psychomotor deficits by targeting frontal dopamine projections

Our work reveals a specific circuit target for long-lasting restoration of cognitive control functions. Dopamine neurons projecting to the frontal cortex are located in the VTA but are distinct from those projecting to the nucleus accumbens (*Björklund and Dunnett, 2007*; *Lammel et al., 2008*). Our initial chemogenetic neuromodulation strategies targeted most of the dopamine neurons in the VTA. While changes in the mesofrontal dopamine circuit were clearly induced, we cannot rule out other potential changes in the brain that might be also induced by such a targeting strategy and result in the lack of net effect related to amphetamine reactivity. The more refined optogenetic (SSFO) neuro-stimulation method allowed us to specifically target the dopaminergic axons projecting to the frontal cortex. Enhancing their activity transiently in adolescence is sufficient to not only restore memory-guided decision-making in the Y-maze task but also prevent hyper-reactivity to amphetamine. Thus, the mesofrontal dopaminergic circuit may provide an important therapeutic target to restore both cognitive control functions and prevent psychomotor symptoms.

We did not examine the degree of bouton growth in the SSFO cohort, which is a limitation of this study. Accurate quantification of dopamine boutons requires the co-injection of another AAV vector encoding synaptophysin-GFP to label the boutons. Because we used light to directly stimulate SSFO-labeled dopaminergic axons in the frontal cortex, we were concerned that co-injecting another AAV vector may dilute SSFO labeling of axons and reduce the efficacy of optogenetic stimulation. Given the behavioral benefits we observed, we would expect an increase in bouton density after optogenetic stimulation. A systematic optimization of viral co-labeling and optogenetic stimulation protocols will facilitate examination of the impact of SSFO stimulation at the structural level in future studies.

This study focused on the M2 region of the frontal cortex because it is functionally required for memory-guided Y-maze navigation, generates behavioral choice-related neural signals during spatial navigation, and is optically most accessible. The medial prefrontal regions (anterior cingulate, prelimbic, and infralimbic) ventral to M2 also receive dense dopaminergic innervation and can act in concert with M2 in decision-making (*Barthas and Kwan, 2017*; *Sul et al., 2011*; *Sul et al., 2010*). As dopaminergic innervations to the frontal cortical regions progress in a ventral-to-dorsal direction during development (*Hoops and Flores, 2017*; *Kalsbeek et al., 1988*), how the changes induced by adolescent dopamine stimulation may proceed spatial-temporally across different frontal subregions requires more extensive investigation in the future.

In conclusion, our results suggest that adolescent frontal dopaminergic projections may provide an effective target to achieve fast-acting and enduring improvement of cognitive function while avoiding psychotic exacerbation, which has been a long-standing challenge for neuroscience and psychiatric research (*Goff et al., 2011*; *Green, 2007*; *Insel, 2010*; *Miyamoto et al., 2005*). Developmentally guided and circuit-based intervention strategies that use clinical brain stimulation, local pharmacological application, or behavioral training methods (*Dahl et al., 2018*; *Larsen and Luna, 2018*; *Lüscher et al., 2015*; *Mukherjee et al., 2019*) to engage the adolescent mesofrontal dopamine circuit may offer new routes to treat cognitive and behavior control disorders.

## Materials and methods

**Key resources table**

| Reagent type (species) or resource | Designation | Source or reference | Identifiers | Additional information |
|---|---|---|---|---|
| Gene (*Mus musculus*) | *Arc* | NCBI Gene ID | 11838 | |
| Gene (*Mus musculus*) | *Disc1* | NCBI Gene ID | 244667 | |

*Continued on next page*

*Continued*

| Reagent type (species) or resource | Designation | Source or reference | Identifiers | Additional information |
|---|---|---|---|---|
| Strain, strain background (*Mus musculus*) | C57BL/6 | The Jackson Laboratory | 000664 | |
| Genetic reagent (*Mus musculus*) | *Arc-/-* | The Jackson Laboratory | 007662 | |
| Genetic reagent (*Mus musculus*) | *Disc1+/-* | The Jackson Laboratory | 036106 | |
| Genetic reagent (*Mus musculus*) | Th-Cre | MMRRC | 031029 | |
| Strain, strain background (Adeno-associated Virus) | AAV2/9 -Syn-GCaMP6s | Boston Children's Hospital Viral Core | | |
| Strain, strain background (Adeno-associated Virus) | AAV8-CaMKIIa-hM4D(Gi)-mCherry | UNC Vector Core | | |
| Strain, strain background (Adeno-associated Virus) | AAV8-Syn-mCherry | Boston Children's Hospital Viral Core | | |
| Strain, strain background (Adeno-associated Virus) | AAV2/9-phSyn1(S)-Flex-tdTomato-T2A-SynEGFP-WPRE | Boston Children's Hospital Viral Core | | |
| Strain, strain background (Adeno-associated Virus) | AAV9.CAG.Flex.tdTomato.WPRE.bGH | Penn Vector Core | | |
| Strain, strain background (Adeno-associated Virus) | pAAV8-hSyn-DIO-hM3D(Gq)-mCherry | UNC Vector Core | | |
| Strain, strain background (Adeno-associated Virus) | pAAV8-hSyn-DIO-mCherry | UNC Vector Core | | |
| Strain, strain background (Adeno-associated Virus) | AAV1.Syn.Flex.GCaMP6s.WPRE.SV40 | Penn Vector Core | | |
| Strain, strain background (Adeno-associated Virus) | pAAV9-EF1a-DIO-hChR2(C128S/D156A)-EYFP | Penn Vector Core | | |
| Strain, strain background (Adeno-associated Virus) | AAV2/1.CAG.FLEX.EGFP.WPRE.bGH | Penn Vector Core | | |
| Chemical compound, drug | d-Amphetamine | Sigma | A5880 | |
| Chemical compound, drug | Clozapine *N*-oxide | Enzo Life Sciences | BML-NS105-0005 | |
| Chemical compound, drug | SCH23390 | Sigma | D054 | |
| Software, algorithm | Matlab | MathWorks | | |
| Software, algorithm | Mosaic | Inscopix | | |
| Software, algorithm | ImageJ | NIH | | |
| Software, algorithm | Prizm | GraphPad | | |

## Experimental model and subject details

*Arc-/-* (*Wang et al., 2006*), *Disc1+/-* (*Kim et al., 2021*), and hemizygote Th-Cre transgenic (*Gong et al., 2007*) mice in the C57BL/6 strain were used in this study. *Arc-/-*;Th-Cre and *Disc1+/-*;Th-Cre lines were generated by in-house breeding. Mice were normally housed in groups of two to five animals. Estrous cycles are known to affect frontal dopamine activity (*Becker and Cha, 1989*; *Dazzi et al., 2007*) and the experiments in this study are limited to male mice of appropriate genotypes. It will be important for future studies to examine female mice at various estrous stages. Litter mates were randomly assigned into control or experimental groups. Behavioral tests were run blinded to the experimental conditions. Experimental protocols were approved by the National Institute of Mental Health Animal Care and Use Committee and the University Committee on Animal Resources (UCAR) at the University of Rochester Medical Center.

## Method details

### Spontaneous alternation in the Y-maze

A three-arm plexiglass Y-maze (45 cm length × 8 cm width × 12 cm height for each arm) was used to test spontaneous alternation. Animals were brought into the testing room at least 30 min prior to the start of the session to acclimate to the room. Room lights were dimmed to ~10–15 lux. Animals were placed in the start arm facing away from the center. Recording of the session was started when the animal started to move toward the center and the animal was allowed to freely explore the Y-maze for 8 min. The Y-maze was cleaned with 70% ethanol between animals.

Videos of the sessions were analyzed blindly offline. Arms were identified as A, B, C. Starting from the start arm, entry into each arm was recorded as a sequence of arm letters. Entry was recorded if all four paws of the animal entered the arm. The number of alternating entries (triplet sequences with non-repeating letters, e.g. ABC) and total entries (total number of letters) were counted. Alternation percentage was calculated as: (total alternating entries/(total entries-2)) × 100.

### Amphetamine-induced locomotion

An animal was placed in an open field arena (50 cm × 50 cm × 25 cm) and monitored by video camera for 10 min. After that, d-amphetamine (1.5 mg/kg) was injected (i.p.), and the animal was monitored by video camera for another 60 min in the arena. Locomotion was automatically tracked using the animal's body position by the Limelight video tracking system (Actimetrics-Coulbourn Instruments).

### Calcium imaging with head-mounted miniature microscope

*WT* and *Arc-/-* animals were prepared for surgery following previous published procedures (*Cao et al., 2015*; *Cao et al., 2013*; *Li et al., 2017*). Mice were anesthetized with Avertin (1.5% solution given at 0.01 ml/g, i.p.) and treated with dexamethasone (0.2 mg/kg, s.c.) and carprofen (5 mg/kg, s.c.) to prevent brain swelling and inflammation. A piece of skull (3.5 mm in diameter) in the frontal cortex was removed after high-speed dental drilling. AAV2/9 -Syn-GCaMP6s ($3–5 \times 10^{12}$ copies/ml, 0.6 µl per animal) was infused into M2 frontal cortex (Bregma, AP1.7, ML0.8) from the pial surface using a micro-syringe pump (*Li et al., 2017*). A 3 mm coverslip was used to seal the cranial window and the exposed scalp was sutured after cranial window surgery. 7–10 days later, the baseplate of a miniaturized integrated fluorescent microscope (Inscopix) was fixed on top of the coverslip. Animals were habituated to the attachment of the microscope (20 min per day for 4 days), then imaged during the spontaneous Y-maze alternation task.

Calcium imaging was performed in freely moving mice using the head-attached microscope (LED power: 0.6–1.0 mW; camera resolution: 1440×1080 pixels). Images were acquired at 30 Hz using nVista 2.0 (Inscopix). At the beginning of each imaging session, the protective cap of the previously implanted baseplate was removed, and the microscope was attached. The imaging field of view was 900 µm×600 µm at 0.65 µm/pixel resolution and the imaging depth was selected by adjusting the focus of the microscope until clear neuronal signals were observed in online ΔF/F calcium video. The focal plane was 150–250 µm below the lens. Mouse behavior was recorded with a video camera (Limelight), which was synchronized with calcium imaging using the trigger-out signal from nVista.

Calcium imaging videos were analyzed by using Mosaic software (Inscopix) and custom-written scripts in Matlab following published algorithms (*Hyvärinen and Oja, 2000*; *Mukamel et al., 2009*). Raw videos were first down-sampled by fourfold along spatial dimensions to reduce file size and noise. The mean fluorescence intensity of each pixel during a recording session (8 min) was calculated as F0 and changes in pixel intensity at time t were expressed as (Ft - F0)/F0 or ΔF/F0. To extract active neuronal signals, principal component and independent component analysis was applied to the spatial-temporal data matrices of ΔF/F0 using CellSort and fastICA toolboxes (these toolboxes are freely downloadable from Matlab central). This analysis decomposes a spatiotemporal data matrix into independent components based on the skewness of data distribution. Each component has a characteristic spatial filter over the imaged area and a corresponding temporal signal during the imaging period. The spatial filter and the temporal signal of each component were graphically displayed and inspected by human observers who were blind to the experimental conditions of each video. If the spatial filter for a component overlapped with the dark shadows casted by blood vessels in the F0 image, this component was likely contributed by blood flow and was therefore rejected. In addition, since calcium signals show characteristic fast-rising and slow-decaying time course (*Chen*

*et al., 2013*), the temporal skewness of calcium signals is expected to be positive and those components with skewness less than 1 were rejected (*Mukamel et al., 2009*). For each selected component, the corresponding temporal signal of each neuron was calculated from the ΔF/F0 video by subtracting the median value of the background area (in a doughnut shape surrounding the target cell) from the average value of the cell soma area.

To identify periods of increased neuronal activity, we searched for the rising phase of each calcium event (peak ΔF/F0 >3 standard deviation of baseline fluctuation), which has been shown closely associated with neuronal spiking activities (*Chen et al., 2013*). The start of this rising phase is detected when the first derivative of ΔF/F0 (calculated in 200 ms moving window) rises above 0 and continues to increase above 5 standard deviation of baseline fluctuation, and the end of this rising phase is detected when the first derivative of ΔF/F0 falls below 0. The magnitude of the calcium event is calculated as the ΔF/F0 difference between the start and end of the event.

To measure an animal's movement in the Y-maze, the center of body mass was tracked in background-subtracted behavioral video using custom-written Matlab scripts. The animal's movement in Y-maze was divided into trials. Each trial began when the animal started moving away from the terminal of one arm and ended after it stopped at the terminal of another arm. To analyze neuronal activity changes along the Y-maze track, the animal's position in each trial was mapped into 20 equally divided location bins from one terminal to another. The calcium activity for each neuron in each trial was then averaged according to the binned position. To compare neuronal activity under different experimental conditions, the spatially averaged activities were further averaged across trials. The resulting traces from all detected cortical neurons were sorted based on their peak activation time and displayed in temporal raster plots. The proportion of neurons that showed maximal activity at each binned location was then calculated. To measure the alternation selectivity of neurons, neuronal activity difference between alternating and non-alternating trials were calculated at each track position. If the difference at a particular position is higher than 2 standard deviations of the differences at all other positions, the activity was determined as alternation-selective at the designated position.

## Frontal cortex inhibition and Y-maze testing

Animals were prepared for surgery following previous published procedures (*Cao et al., 2015*; *Cao et al., 2013*; *Li et al., 2017*). Mice were anesthetized with Avertin (1.5% solution given at 0.01 ml/g, i.p.). 0.7 µl of AAV8-CaMKIIa-hM4D(Gi)-mCherry (4×10^12 copies/ml, UNC Vector Core) or AAV8-Syn-mCherry (3×10^12 copies/ml, Boston Children's Hospital Viral Core) was injected into the M2 frontal cortex (from bregma: AP 1.7, ML 0.5, DV 1.0 mm) bilaterally. After 2 weeks, mCherry-expressing controls were injected with CNO (3 mg/kg) and Gi-expressing animals were injected with either CNO (3 mg/kg) or 0.9% saline control. Animals were run in the Y-maze 1 hr after the injections. After behavioral testing, animals were perfused and M2 frontal cortical expression was confirmed by confocal microscopy of brain sections.

## Labeling and imaging of VTA dopaminergic neurons and projections to the frontal cortex

AAV2/9-phSyn1(S)-Flex-tdTomato-T2A-SynEGFP-WPRE (4×10^13 copies/ml, Boston Children's Hospital Viral Core, 0.5 µl) and AAV9.CAG.Flex.tdTomato.WPRE.bGH (9×10^12 copies/ml, Penn Vector Core, 0.3 µl) were mixed together and injected into the right hemisphere of the midbrain region (from bregma: AP –3.2, ML 0.5, DV 4.4mm) of *Arc+/+;Th-Cre* and *Arc-/-;Th-Cre* animals. Animals were allowed to recover for ~2 weeks. Animals were then perfused with 0.9% saline followed by 4% paraformaldehyde (PFA) and post-fixed overnight at 4°C. Coronal 100 µm thick sections in the frontal cortical and midbrain regions were prepared with a sliding microtome. Three sections in the frontal cortical region (AP 1.6, 1.8, and 2.0; 25× lens, 10 µm Z-stack, 10 stacks per section) and five sections in the midbrain region (AP –2.9, –3.1, –3.3, –3.5, and –3.7; 10× lens, single-frame images) were imaged in both the green and red channels using confocal microscopy (Olympus FV1000). For the frontal cortex, each image stack was maximum projected, and the 10 stacks were stitched together for each section. Animals that did not show VTA labeling were excluded from further analysis.

Image analysis was carried out using custom Matlab scripts. For boutons and axons, an ROI for the M2 region was drawn in each section using the mouse brain atlas (Paxinos) as reference, and boutons and axons were identified within this region. SypGFP-labeled green channel images were used for

bouton quantification. Boutons were detected automatically using a Laplacian filter and thresholded at 5 standard deviations above background. tdTomato-labeled red channel images were used for the axon quantification. Axons were detected automatically using a Hessian filter, thresholded at 2 standard deviations above background, and skeletonized. For the midbrain sections, cells were detected automatically using a Laplacian filter and thresholded at 2 standard deviations above background. For each animal, the bouton density was normalized by the total axon length, and the total axon length was normalized by the total number of tdTomato-labeled midbrain dopamine cells.

## Expression of DREADD-Gq in midbrain dopamine neurons and validation of CNO-induced mesofrontal activation

0.7 µl of pAAV8-hSyn-DIO-hM3D(Gq)-mCherry ($6\times10^{12}$ copies/ml, UNC Vector Core) or pAAV8-hSyn-DIO-mCherry ($3\times10^{12}$ copies/ml, UNC Vector Core) was injected into the right hemisphere of the midbrain region (from bregma: AP –3.2, ML 0.5, DV 4.4 mm) and 0.7 µl of AAV1.Syn.Flex.GCaMP6s.WPRE.SV40 ($2\times10^{12}$ copies /ml, Penn Vector Core) was injected into the right hemisphere of the frontal cortex (from bregma: AP 1.7, ML 0.7, DV 0.4 mm) of Th-Cre adult animals.

After ~2 weeks, a cranial window was opened above the AAV-GCaMP6 injected region in the frontal cortex (from Bregma: AP 1.0–3.0 mm, ML 0.3–1.3 mm, covering the M2 region) in animals anesthetized with isoflurane (~1.5%). The cranial window was filled with silicone gel, covered with a glass coverslip, and sealed with dental cement. A head plate was glued on the skull for fixation during imaging. The animals were then taken off the anesthesia and allowed to recover for ~1 hr before imaging. A two-photon microscope (FV1000, Olympus) was used to image the brain under the cranial window (excitation laser: 900 nm) using a 20× water immersion lens (NA 0.95) in the head fixed awake animal. Animals were imaged before and 1 hr after CNO (1 mg/kg, i.p.) injection. After imaging, animals were perfused, and DREADD-Gq expression was confirmed in the midbrain regions. Animals that did not show VTA labeling were excluded from further analysis.

Images were analyzed using NIH ImageJ. Two 2 min movies of spontaneous activity before and after CNO were analyzed. The mean pixel intensity in each image frame of the movie was calculated as Ft. Baseline fluorescence (**F0**) was defined as the average of the fluorescent signals (Ft) in the time series. Changes in calcium signals (ΔF/F) are calculated as (Ft-F0)/F0. The standard deviation of the (ΔF/F) was used as a quantitative measure of overall cortical activity.

## CNO stimulation of DREADD-Gq in midbrain dopamine neurons and behavior testing

Animals were prepared for surgery as above. 0.7 µl of pAAV8-hSyn-DIO-hM3D(Gq)-mCherry or pAAV8-hSyn-DIO-mCherry was bilaterally injected into the midbrain region (from bregma: AP –3.2, ML 0.5, DV 4.4mm) of *Arc-/-*;Th-Cre or *Disc1+/-*;Th-Cre animals within postnatal days P21–P25. CNO (1 mg/kg, i.p.) was injected 1 time per day for 3 days into the animals within postnatal days P35–P42. For 1-day-after tests, animals were tested on the Y-maze 1 day after the last injection day. For adulthood tests, animals were tested at 2–3 month age on the Y-maze. Some of the adult animals were also tested for amphetamine-induced locomotion as described above. After behavior testing, animals were perfused, and DREADD-Gq expression was confirmed in the ventral midbrain region. Animals that did not show VTA labeling were excluded from further analysis.

## CNO stimulation of DREADD-Gq in midbrain dopamine neurons and in vivo imaging of cortical activity

Animals were prepared for surgery as described above. 0.7 µl of pAAV8-hSyn-DIO-hM3D(Gq)-mCherry or pAAV8-hSyn-DIO-mCherry was bilaterally injected into the midbrain region (from bregma: AP –3.2, ML 0.5, DV 4.4mm) of *Arc-/-*;Th-Cre animals within postnatal days P21–P25. CNO (1 mg/kg, i.p.) was injected 1 time daily for 3 days into the animals for activation within postnatal days P35–P42. At adulthood, GCaMP6 labeling, cranial window opening, and imaging during Y-maze were carried out as described above. After an interval of at least 1 day after miniscope imaging, animals were used for VTA electrical stimulation coupled with in vivo two-photon imaging. A bipolar stimulation electrode was placed into the VTA (from bregma: AP –3.2, ML 0.5, and DV 4.5 mm) and glued in place in animals anesthetized with isoflurane (~1.5%). A head plate was also glued on to the skull. The animals were

then taken off the anesthesia and allowed to recover for ~1 hr before imaging. Time series images lasting ~40 s (115 frames at 0.351 s/frame) were taken for each stimulus train, with the VTA stimulus delivered at 20 s after the start of imaging. Image analysis was carried out as described above. Baseline fluorescence (F0) was defined as the average of the fluorescent signals (Ft) in the first 15 s of the time series. Changes in calcium signals (ΔF/F) are calculated as (Ft-F0)/F0. After imaging, animals were perfused, and DREADD-Gq expression was confirmed in the midbrain regions. Animals that did not show VTA labeling were excluded from further analysis.

## CNO stimulation of DREADD-Gq in midbrain dopamine neurons and imaging of mesofrontal axons and boutons

Animals were prepared for surgery as described above. A mixture of 0.4 µl of pAAV8-hSyn-DIO-hM3D(Gq)-mCherry or pAAV8-hSyn-DIO-mCherry 0.5 µl AAV2/9-phSyn1(S)-Flex-tdTomato-T2A-SynEGFP-WPRE and 0.3 µl AAV9.CAG.Flex.tdTomato.WPRE.bGH was injected into the midbrain region (from bregma: AP –3.2, ML 0.5, DV 4.4 mm) of *Arc-/-;Th-Cre* animals within postnatal days P21–P25. The procedure was later modified with a 0.6 µl of pAAV8-hSyn-DIO-hM3D(Gq)-mCherry or pAAV8-hSyn-DIO-mCherry first and then immediately followed by an injection of a mixture of 0.5 µl AAV2/9-phSyn1(S)-Flex-tdTomato-T2A-SynEGFP-WPRE and 0.3 µl AAV9.CAG.Flex.tdTomato.WPRE.bGH. 12 animals from the first procedure and 8 animals from the latter procedure were pooled together for the analysis. CNO (1 mg/kg, i.p.) was injected one time daily for 3 days into the animals within postnatal days P35–P42. At adulthood, animals were then perfused with 0.9% saline followed by 4% PFA and post-fixed overnight at 4°C. Coronal 100 µm thick sections in the frontal cortical and midbrain regions were prepared with a sliding microtome. Three sections in the frontal cortical region (AP 1.6, 1.8, and 2.0) and five sections in the midbrain region (AP –2.9, –3.1, –3.3, –3.5, and –3.7) were imaged in both the green and red channels using confocal microscopy. Image analysis was carried out as described above.

## SSFO expression in midbrain dopamine neurons and validation of cortical light activation

0.7 µl of pAAV9-EF1a-DIO-hChR2(C128S/D156A)-EYFP (titer $5×10^{13}$ copies/ml diluted 1:20 in PBS, Penn Vector Core) or AAV2/1.CAG.FLEX.EGFP.WPRE.bGH (titer $8×10^{12}$ copies/ml, Penn Vector Core) was injected into the midbrain region (from bregma: AP –3.2, ML 0.5, DV 4.4 mm) and 0.7 µl of AAV1.Syn.Flex.GCaMP6s.WPRE.SV40 (titer $8×10^{12}$ copies/ml, Penn Vector Core) was injected into the frontal cortex (from bregma: AP 1.7, ML 0.7, DV 0.4 mm) of Th-Cre adult animals.

After ~2 weeks, cranial window (from Bregma: AP 1.0–2.5 mm, ML 1 mm) opening and two-photon imaging was carried out as described above. Animals were imaged before and 30 min after light activation with an optical fiber (200 µm in diameter, Thor Labs) connected to a 473 nm solid-state laser diode (CrystaLaser) with ~10 mW output from the fiber. Three spots separated by around 200 µm anterior-posterior in the center of the window were activated with a 2 s light pulse for each spot. After imaging, animals were perfused, and SSFO expression was confirmed in the midbrain regions. Images were analyzed as described above. Three min times series images of spontaneous activity before and after CNO were analyzed. Baseline fluorescence (F0) was defined as the average of the fluorescent signals (Ft) in the time series. The standard deviation of the (ΔF/F) was used as a quantitative measure of overall cortical activity.

## Light activation of SSFO in frontal dopaminergic axons and behavioral testing

0.7 µl of pAAV9-EF1a-DIO hChR2(C128S/D156A)-EYFP or AAV2/1.CAG.FLEX. EGFP.WPRE.bGH was bilaterally injected into the midbrain region (from bregma: AP –3.2, ML 0.5, DV 4.4mm) of *Arc-/-;Th-Cre* animals within postnatal days P21–P25. Around postnatal days P35–P42, a cranial window was opened in above the frontal cortex (from Bregma: AP 1.0–2.5 mm, ML 1 mm across both sides of the midline) in animals anesthetized with isoflurane (~1.5%). The cranial window was filled with silicone gel, covered with a glass coverslip, and sealed with dental cement. A head bar was glued on the skull for fixation during light activation. Animals were allowed to wake and recover for at least 2 hr. Animals were then head fixed and an optical fiber (200 µm in diameter, Thor Labs) connected to a 473 nm

solid-state laser diode (CrystaLaser) with ~10 mW output from the fiber was used to deliver 2 s light pulses to the frontal cortex. Three spots separated by around 200 µm anterior-posterior in the center of the window were activated in each hemisphere. The light activation was repeated for 2 more days. For 1-day-after tests, animals were tested in the Y-maze 1 day after the last injection day. For adulthood tests, animals of 2–3 month age were first tested in the Y-maze, then tested for amphetamine-induced locomotion as described above. After behavior testing, animals were perfused, and SSFO expression was confirmed in the midbrain regions. Animals that did not show VTA labeling were excluded from further analysis.

## Quantification and statistical analysis

Statistical analyses were performed in Prism 9 or Matlab. No statistical methods were used to predetermine sample size. Sample sizes were chosen based on previous studies using similar techniques to determine biological effects (*Liu et al., 2018*; *Managò et al., 2016*; *Mastwal et al., 2014*; *Wang et al., 2017*). Statistical differences between the means of two groups were evaluated with two-tailed *t*-test and normality was confirmed with Shapiro-Wilk test. Statistical differences between the means of multiple groups were determined using ANOVA. Statistical differences between two proportions were evaluated with chi-squared test.

## Acknowledgements

The authors thank Drs. Y Chudasama, F Wang, Z He for critical reading of the manuscript and J Pulizzi for technical assistance. This work was supported by grants from National Institutes of Health (ZIAMH002897 and R01MH127737 to KHW, F32MH124298 to RS, U19MH106434 and R35NS116843 to HS, and R35NS097370 to GLM) and Del Monte Institute for Neuroscience at University of Rochester (to KHW).

## Additional information

### Funding

| Funder | Grant reference number | Author |
| --- | --- | --- |
| National Institutes of Health | ZIAMH002897 | Kuan Hong Wang |
| University of Rochester | Del Monte Institute for Neuroscience | Kuan Hong Wang |
| National Institutes of Health | F32MH124298 | Rianne Stowell |
| National Institutes of Health | U19MH106434 | Hongjun Song |
| National Institutes of Health | R35NS097370 | Guo-Li Ming |
| National Institutes of Health | R01MH127737 | Kuan Hong Wang |

The funders had no role in study design, data collection and interpretation, or the decision to submit the work for publication.

### Author contributions

Surjeet Mastwal, Xinjian Li, Data curation, Formal analysis, Investigation, Visualization, Methodology, Writing - original draft, Writing - review and editing; Rianne Stowell, Data curation, Formal analysis, Validation, Investigation, Visualization, Methodology, Writing - original draft, Writing - review and editing; Matthew Manion, Methodology, Writing - review and editing; Wenyu Zhang, Investigation, Methodology, Writing - review and editing; Nam-Shik Kim, Ki-Jun Yoon, Hongjun Song, Guo-Li Ming, Resources, Writing - review and editing; Kuan Hong Wang, Conceptualization, Data curation, Formal

analysis, Supervision, Funding acquisition, Visualization, Writing - original draft, Project administration, Writing - review and editing

**Author ORCIDs**
Ki-Jun Yoon (ID) http://orcid.org/0000-0003-2985-2541
Kuan Hong Wang (ID) http://orcid.org/0000-0002-2249-5417

**Ethics**
This study was performed in strict accordance with the recommendations in the Guide for the Care and Use of Laboratory Animals of the National Institutes of Health. Experimental protocols were approved by the National Institute of Mental Health Animal Care and Use Committee (Protocol Number: GCP-01) and the University Committee on Animal Resources (UCAR) at the University of Rochester Medical Center (Protocol Number: 102193). All surgery was performed under isoflurane anesthesia, and every effort was made to minimize suffering.

Joint Public Review: https://doi.org/10.7554/eLife.87414.3.sa1
Author Response https://doi.org/10.7554/eLife.87414.3.sa2

---

# Additional files

## Supplementary files
• MDAR checklist

## Data availability
All data generated or analyzed during this study are included in the manuscript and supporting file. Source data files have been provided containing the numerical data used to generate the figures.

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
