## [Editor Report · eLife assessment]

This is an **important** study that addresses the interesting question of whether stimulation of DA input to prefrontal cortex during adolescence can be used to rescue genetic defects on DA regulation of PFC function. The conclusions are **convincingly** supported by the data together with discussion of some limitations of the approach. This story will of interest to a broad group of neuroscientists interested in regulation of prefrontal cortical function in behavior.

---

## [Referee Report · Joint Public Review]

The authors have previously established that activation of dopamine inputs to prefrontal cortex during adolescence can drive increases in mPFC DA bouton number and enhanced mPFC activity in WT mice. The current study was designed to test the hypothesis that neural circuit plasticity during adolescence can be targeted to restore cortical function under conditions of developmental disruptions that are relevant to psychiatric disorders.

Specifically, the manuscript explores how transient adolescent stimulation of ventral midbrain neurons that project to the frontal cortex may help to improve performance on certain memory tasks. The authors used DREADDs to regulate the mesofrontal cortical dopamine system in two mouse models - one with a reporter replacing the Arc gene, and another with knockout of the schizophrenia-associated gene Disc1, both of which are thought to have reduced prefrontal cortical activity. The manuscript provides an interesting set of observations that DREADD-based activation over only 3 days during adolescence provides a fast-acting and long-lasting improvement in performance on Y-maze spontaneous alternation as well as aspects of neuronal function as assessed using in vivo imaging methods.

A strength of this study is that the authors performed key manipulations using age and dose/intensity as dependent variables to show that the level of neural circuit activation during adolescence follows an inverted U-shape pattern, though the precise postsynaptic mechanisms underlying the positive impact of adolescent mesofrontal dopamine neuron stimulation were not addressed.

One limitation discussed by the reviewers is that using TH-Cre mice (as compared with DAT-Cre) to drive transgene expression in VTA neurons could lead to expression outside the dopaminergic population of neurons, though in the revision the authors have provided additional lines of evidence to support their model of dopamine regulation of frontal cortex in this study.

Collectively, this is a well-design study with many strengths and novel findings that are likely to positively impact a widespread of disciplines within the biological psychiatry and neuroscience field.

---

## [Author Response]

The following is the authors’ response to the original reviews.

First and foremost, we would like to thank all the editors and reviewers for their thoughtful and thorough evaluations of our manuscript. We greatly appreciate their assessment about the novelty and strength in this study and have revised the manuscript according to their recommendations. Below are our detailed responses and revisions based on the reviewer recommendations.

**Reviewer #1 (Recommendations For The Authors):**
1. It is unclear the rationale for choosing the P35-42 adolescent window for stimulating the mesofrontal dopamine system.

The dopaminergic innervation in the mesofrontal circuit exhibits a protracted maturation from P21 to P56 (Kalsbeek, Voorn et al. 1988, Niwa, Kamiya et al. 2010, Naneix, Marchand et al. 2012, Hoops and Flores 2017). P35-42 is in the center of this period and captures the mid-adolescent stage in rodents (Spear 2000). We have previously shown that increasing dopamine neuron activity by wheel running or optogenetic stimulation during this period, but not adulthood, can induce formation of mesofrontal dopaminergic boutons and enhance mesofrontal circuit activity in wild-type mice (Mastwal, Ye et al. 2014). We therefore chose the P35-P42 adolescent window to stimulate the mesofrontal dopamine circuit and test the long-term effect of this intervention on the frontal circuit and memory-guided decision-making deficits in mutant mice. We have detailed this rationale in the revised manuscript when we first introduced this intervention.

2). Please provide a justification for choosing the optical recording M2 neuronal activity instead of the prelimbic prefrontal cortex, which has been known to show the highest levels of dopamine terminals.

While the prelimbic area has the highest level of dopamine terminals among frontal cortical regions, a robust presence of dopaminergic terminals and dopamine release in the M2 frontal cortex have been well documented (Berger, Gaspar et al. 1991, Mastwal, Ye et al. 2014, Aransay, Rodriguez-Lopez et al. 2015, Patriarchi, Cho et al. 2018). The M2 cortex plays an important role in action planning, generating the earliest neural signals among frontal cortical regions that are related to upcoming choice during spatial navigation (Sul, Kim et al. 2010, Sul, Jo et al. 2011). Our chemogenetic inactivation experiments (Supplementary Fig 1) has further confirmed the involvement of M2 in the memory-guided Y-maze navigation task used in this study. Technically, M2 has the advantage of being more amendable to optical recording of neuronal activity without the tissue damage caused by implanting a lens, which would be necessary for deeper areas such as the prelimbic cortex. We have provided this justification in the revised manuscript.

3). What was the rationale for using the 3-day chemogenetic stimulation paradigm?

Our previous work in wild-type adolescent mice showed that a single optogenetic stimulation session or a 2-hr wheel running session is sufficient to induce bouton formation in mesofrontal dopaminergic axons (Mastwal, Ye et al. 2014). In this study, we sought to rescue existing structural and functional deficits in the mesofrontal dopaminergic circuits due to genetic mutations. Because previous studies suggested that an optimal level of dopamine is important for normal cognitive function (Arnsten, Cai et al. 1994, Robbins 2000, Floresco 2013), we elected to do multiple stimulation sessions to boost the potential rescue effects. We tested both a 3-day and a 3-week stimulation paradigm, and found that the 3-day, but not the 3-week paradigm led to robust functional improvement (Fig. 5). These results indicate that moderate but not excessive stimulation of dopamine neurons can provide functional improvement of a deficient mesofrontal circuit. We have revised our text to clarify the rationale for these experiments.

4). A major maturational event occurring in the prefrontal cortex is the gain of local GABAergic transmission, which is crucial for sustaining proper levels of Y-maze tasks. I am wondering if the authors have any thoughts about what is really happening at the postsynaptic level following adolescent dopamine stimulation.

The developmental increases in dopaminergic innervation to the frontal cortex and local GABAergic transmission are likely synergistic processes, which both contribute to the maturation of high-order cognitive functions supported by the frontal cortex (Caballero and Tseng 2016, Larsen and Luna 2018). Previous electrophysiological studies have suggested that dopamine can act on five different receptors expressed in both excitatory and inhibitory postsynaptic neurons (Seamans and Yang 2004, Tseng and O'Donnell 2007, O'Donnell 2010). At the network level, dopaminergic signaling can increase the signal-to-noise ratio and temporal synchrony of neural activity during cognitive tasks (Rolls, Loh et al. 2008, Vander Weele, Siciliano et al. 2018, Lohani, Martig et al. 2019). As the frontal GABAergic inhibitory network undergoes major functional remodeling during adolescence (Caballero and Tseng 2016), adolescent stimulation of dopamine neurons may interact with this maturational process to promote a network configuration conducive for synchronous and high signal-to-noise neural computation (Porter, Rizzo et al. 1999, Murty, Calabro et al. 2016, Mukherjee, Carvalho et al. 2019). The microcircuit mechanisms underlying adolescent dopamine stimulation induced changes, particularly in the GABAergic inhibitory neurons, will be an exciting direction for future research. We have extended our discussion about these points in the revised manuscript.

5). A change in the density of dopamine boutons is unlikely to be limited to the M2 region in Arc-/- mice. The authors should provide some data illustrating that similar changes are widespread across the medial prefrontal cortex, and that the optical recording in the M2 region was preferred for technical limitations and to avoid damaging areas in the frontal cortex.

As discussed above, this study focused on the M2 region of the frontal cortex because it is functionally required for memory-guided Y-maze navigation, generates behavioral choice-related neural signals during spatial navigation, and is optically most accessible. The medial prefrontal regions (anterior cingulate, prelimbic and infralimbic) ventral to M2 also receive dense dopaminergic innervation and can act in concert with M2 in decision making (Sul, Kim et al. 2010, Sul, Jo et al. 2011, Barthas and Kwan 2017). As dopaminergic innervations to the frontal cortical regions progress in a ventral-to-dorsal direction during development (Kalsbeek, Voorn et al. 1988, Hoops and Flores 2017), how the changes induced by adolescent dopamine stimulation may proceed spatial-temporally across different frontal subregions requires more extensive investigation in the future. We have added this discussion into the revised manuscript.

**Reviewer #2 (Public Review):**
The manuscript by Mastwal and colleagues explores how transient adolescent stimulation of ventral midbrain neurons that project to the frontal cortex may help to improve performance on certain memory tasks. The manuscript provides an interesting set of observations that DREADD-based activation over only 3 days during adolescence provides a fast-acting and long-lasting improvement in performance on Y-maze spontaneous alternation as well as aspects of neuronal function as assessed using in vivo imaging methods. While interesting, there are several weaknesses. First and foremost, it is not clear that the effects the authors are observing are mediated by dopamine. It has been clearly documented that the DAT-Cre line provides a better representation of midbrain dopamine cells in the mouse, particularly near the midline of the ventral midbrain (Lammel et al., Neuron 2015). This is precisely where the cells that project to the frontal cortex are located. Therefore, the selection of TH-Cre is problematic. It is very likely that the authors are labeling a substantial number of non-dopaminergic cells.

We agree with Review 2 that the DAT-Cre line can provide specific labeling of midbrain dopamine neurons, particularly those projecting to the striatum, as discussed in the cited study (Lammel, Steinberg et al. 2015). DAT transports the extracellularly released dopamine back into presynaptic terminals, but it is not essential for dopamine synthesis and release (Sulzer, Cragg et al. 2016). Mesocortical dopamine neurons in the ventral tegmental area (VTA) express very little DAT (Sesack, Hawrylak et al. 1998, Lammel, Hetzel et al. 2008, Li, Qi et al. 2013), which limits the use of the DAT-Cre line to target these neurons (Lammel, Steinberg et al. 2015). Because mesocortical dopamine neurons have strong expression of TH, a key enzyme involved in dopamine synthesis, TH-Cre lines have been extensively used to study the mesocortical pathway (Lammel, Lim et al. 2012, Gunaydin, Grosenick et al. 2014, Ellwood, Patel et al. 2017, Vander Weele, Siciliano et al. 2018, Lohani, Martig et al. 2019). We provide more details below about our rationales for using TH-Cre rather than DAT-Cre mice in our study and the revisions we made in response to the reviewer’s specific recommendations.

**Reviewer #2 (Recommendations For The Authors):**
1). The authors should rigorously demonstrate that there is a reasonable midbrain DA projection to the coordinates that they are assessing and that their effects are due to DA release from these cells. It is not clear that there is a VTA dopaminergic projection to M2 - it does not appear for example in the Allen Mouse Brain Connectivity Atlas (https://connectivity.brainmap.org/projection/experiment/siv/160540751? imageId=160541123&imageType=TWO_PHOTON,SEGMENTATION&initImage=TWO_PHOTON&x=17321&y=15284&z=3). Though there is a projection to the mPFC, at the coordinates the authors report, there does not appear to be any signal from DAT-Cre mice. However, there is much more signal when expression is not restricted to dopamine cells (https://connectivity.brain-map.org/projection/experiment/siv/165975096? imageId=165975158&imageType=TWO_PHOTON,SEGMENTATION&initImage=TWO_PHOTON&x=17950&y=11504&z=3). The argument that these cells may express less TH is not relevant for this particular issue. Therefore, it is possible that the vast majority of observed effects are not in fact mediated by dopamine but another neurotransmitter such as glutamate. While the experiment using SCH23390 does suggest DA receptors may be involved, this result in isolation doesn't alleviate this caveat - there can be, for example, DA release from NE cells (e.g., Takeuchi et al., Nature 2016). While this does not entirely invalidate the authors' results, as their effects of stimulation of ventral midbrain cells to the forebrain don't necessarily have to occur via dopamine - the mechanism by how this is occurring needs to be clear.

While the prelimbic area has the highest level of dopaminergic terminals among frontal cortical regions, a robust presence of midbrain dopaminergic projections and dopamine release in the M2 frontal cortex have been well established by immunostaining, viral labeling, single-cell axon-tracing, and in vivo imaging of recently developed dopamine biosensors (Berger, Gaspar et al. 1991, Mastwal, Ye et al. 2014, Aransay, Rodriguez-Lopez et al. 2015, Ye, Mastwal et al. 2017, Patriarchi, Cho et al. 2018). It has also been reported repeatedly that mesocortical dopamine neurons in the VTA express very little DAT, which is different from mesostriatal dopamine neurons (Sesack, Hawrylak et al. 1998, Lammel, Hetzel et al. 2008, Li, Qi et al. 2013). This limitation in the use of the DAT-Cre line to target mesocortical dopamine neurons has been acknowledged in previous studies (Lammel, Steinberg et al. 2015) and is consistent with the reviewer’s observation of DAT-Cre labeling in the Allen Brain Mouse Connectivity atlas. Additionally, and interestingly, recent extensive evaluation of the DAT-Cre line reported ectopic labeling of multiple non-dopaminergic neuronal populations (Soden, Miller et al. 2016, Stagkourakis, Spigolon et al. 2018, Papathanou, Dumas et al. 2019). Our own evaluation of the DAT-Cre line’s utility for cortical imaging also revealed sparse axonal labeling and sporadic ectopic labeling of cortical cell somas. We have included representative DAT-Cre images in Author response image 1 to highlight the limitations of this line in the study of the dopaminergic mesocortical circuit.

**Author response image 1. sa2fig1:** Example images from DAT-Cre/Ai14 mice. Left most panel shows little axonal labeling in Layer 5/6 of M2. The center panel shows sparse axonal label in Layer 1/2 of M2, but also ectopic labeling of cell soma. The right panel shows a lack of labeling in L1/2 of prelimbic cortex as well. Scale bars 50um.

We as well as others have confirmed that TH immunoreactivity in the frontal cortex can label dopaminergic axons originated from the VTA, and ablation of VTA dopaminergic neurons removes this labeling (Niwa, Jaaro-Peled et al. 2013, Ye, Mastwal et al. 2017). Because mesocortical dopamine neurons have much stronger TH expression than DAT expression (Sesack, Hawrylak et al. 1998, Lammel, Hetzel et al. 2008, Li, Qi et al. 2013, Lammel, Steinberg et al. 2015), TH-Cre lines have been frequently used to label these neurons and study the mesocortical pathway (Lammel, Lim et al. 2012, Gunaydin, Grosenick et al. 2014, Ellwood, Patel et al. 2017, Vander Weele, Siciliano et al. 2018, Lohani, Martig et al. 2019). While TH-Cre expression itself is not restricted to dopaminergic neurons, we targeted our viral injections to the VTA and optogenetic stimulation to the cortical dopaminergic projection target area in M2 (Patriarchi, Cho et al. 2018) to specifically modulate mesofrontal dopaminergic axons. In addition, we tested D1 antagonist’s effects in our manipulations. Although we targeted dopamine neurons in our adolescent stimulation, the final behavioral outcome likely includes contributions from co-released neurotransmitters such as glutamate and non-dopaminergic neurons via network effects (Morales and Margolis 2017, Lohani, Martig et al. 2019), which will be interesting directions for future research. We have revised our results and discussion sections to highlight our rationales for using the TH-Cre line and the open mechanistic questions for future studies.

1. SSFOs don't increase excitability like DREADDs, but rather, cause long-lasting hyperactivity through continuous passage of cations. What the actual firing properties are of these cells over a long period of time is not clear.

We did not measure the precise firing patterns of the dopaminergic neurons targeted by SSFOs but evaluated the effects of SSFO activation on the frontal cortex. Similar to our DREADD-Gq mediated activity changes in the mesofrontal circuit, we found increased frontal cortical activity post-light stimulation of frontal dopamine axons in our SSFO treated animals (Fig 6a-c, S6e). While quantitatively the firing patterns of DREADD-Gq and SSFO activated dopaminergic neurons likely differ, qualitatively both of these manipulations lead to increased mesofrontal circuit activity and improvements in cognitive behaviors. In our previous work with wild-type adolescent mice, both wheel running and a single 10-min session of phasic optogenetic stimulation of the VTA resulted in dopaminergic bouton outgrowth in the frontal cortex (Mastwal, Ye et al. 2014). Taken together, these results suggest that adolescent dopaminergic mesofrontal projections are highly responsive to neural activity changes and a variety of adolescent stimulation paradigms are sufficient to elicit lasting changes in this circuit. We have added this discussion of the limitations and implications of our study into the revised manuscript.

1. It is not clear what the increase in boutons means, given that DA release is thought to largely occur via non-synaptic release.

Although many of dopamine boutons are not associated with defined postsynaptic structures, these axonal boutons and the active zones they contain are the major release sites for dopamine (Goldman-Rakic, Leranth et al. 1989, Arbuthnott and Wickens 2007, Sulzer, Cragg et al. 2016, Liu, Goel et al. 2021). Past studies have established a consistent association between increased dopaminergic innervation in the frontal cortex and an increase in dopamine levels (Niwa, Kamiya et al. 2010, Naneix, Marchand et al. 2012). Our previous work also found that increasing dopaminergic boutons through adolescent VTA stimulation led to prolonged frontal local field potential responses with high-frequency oscillations (Mastwal, Ye et al. 2014), which is characteristic of increased dopaminergic signaling (Lewis and O'Donnell 2000, Gireesh and Plenz 2008, Wood, Kim et al. 2012, Lohani, Martig et al. 2019). Importantly, in our quantification of the structural changes in this study, we evaluated boutons which were labeled with synaptophysin, a molecular marker indicating the presence of synaptic vesicle release machinery (Li, Tasic et al. 2010, Oh, Harris et al. 2014). Thus, our study, taken in the context of the previous work, suggests the increased number of boutons signifying an increase in dopaminergic signaling within the mesofrontal circuit. We have added this discussion into the revised manuscript.

1. The use of Arc and DISC mutants as models of schizophrenia is perhaps a bit overstated - while deficits in prefrontal innervation certainly occur, there are many differences between these models and the human disease states. Language should be toned down accordingly, particularly in the introduction.

We strived to avoid overstating the extent to which the mouse lines are models for specific diseases, but we can appreciate that this may not have been clear in our original writing. We have adjusted our language to better distinguish between the utility of the animal models for the purposes of our study and their relationship to specific human disease states. Particularly in the introduction, we stated that: “Genetic disruptions of several genes involved in synaptic functions related to psychiatric disorders, such as Arc and DISC1, lead to hypoactive mesofrontal dopaminergic input in mice (Niwa, Kamiya et al. 2010, Niwa, Jaaro-Peled et al. 2013, Fromer, Pocklington et al. 2014, Purcell, Moran et al. 2014, Wen, Nguyen et al. 2014, Manago, Mereu et al. 2016). Although there are many differences between these mouse lines and specific human disease states, these mice offer opportunities to test whether genetic deficits in frontal cortex function can be reversed through circuit interventions.”

1. Some experiments are missing proper controls, e.g., Figure 3g-I where a WT mouse should be used as a positive control.

The goal of this experimental design (Fig 3g-i) was to evaluate the potential effects of chemogenetic VTA stimulation in the Arc-/- mice. We used Arc-/- mice with mCherry injections to control for the potential effects of CNO administration. While WT mice could be used to determine if adolescent VTA stimulation would lead to long-lasting enhancement of VTA-to-Cortical transmission, this wouldn’t necessarily be a positive control for our experiments, but rather a separate line of inquiry. As dopamine’s effects often display an inverted-U dose-response curve (Vijayraghavan, Wang et al. 2007, Floresco 2013), evaluating the effects adolescent VTA stimulation in the absence of underlying dopamine deficiency could be an interesting future research direction. We have added this discussion into the revised manuscript.

**Reviewer #3 (Recommendations For The Authors):**
1. Did the SSFO stimulation of the TH+ axons in PFC during adolescence lead to the same long-term change in DA bouton number the authors saw with DREADDs?

We did not examine the degree of bouton growth in the SSFO cohort, which is a limitation of this study. Accurate quantification of dopamine boutons requires the co-injection of another AAV vector encoding Synaptophysin-GFP to label the boutons. Because we used light to directly stimulate SSFO-labeled dopaminergic axons in the frontal cortex, we were concerned that co-injecting another AAV vector may dilute SSFO-labeling of axons and reduce the efficacy of optogenetic stimulation. Given the behavioral benefits we observed, we would expect an increase in bouton density after optogenetic stimulation. A systematic optimization of viral co-labeling and optogenetic stimulation protocols will facilitate examination of the impact of SSFO stimulation at the structural level in future studies. We have added a discussion of the limitation of this study in the revised manuscript.

1. The DISC1 section is far less detailed than the Arc section, and it was not completely clear to me that the mechanisms of dysfunction and rescue were the same in these mice compared with the Arc mice. For example, there was no mention of DA bouton density or the patterned firing of the PFC neurons at the time of decision making.

The initial motivation of this study was to test if adolescent dopamine stimulation can rescue the deficits in the mesofrontal dopaminergic circuit and cognitive function of Arc-/- mice, which were identified in our previous studies (Manago, Mereu et al. 2016). We first conducted multiple levels of analyses including viral tracing, in vivo calcium imaging, and behavioral tests to establish the coherent impacts of adolescent dopamine neuron stimulation on circuits and behaviors. We then examined a range of stimulation protocols to assess the efficacy requirements for cognitive improvement, which is our primary goal. Finally, we included DISC1 mice in our study to test if adolescent dopamine stimulation can also reverse the cognitive deficit in another genetic model for mesofrontal dopamine deficiency. By demonstrating a similar cognitive recuse effect of adolescent VTA stimulation in an independent mouse model, this study provides a foundation for future research to compare the detailed cellular mechanisms that underlie the functional rescue in different genetic models. We have added the discussion of the scope and limitation of this study to the revised manuscript.

References

Aransay, A., C. Rodriguez-Lopez, M. Garcia-Amado, F. Clasca and L. Prensa (2015). "Long-range projection neurons of the mouse ventral tegmental area: a single-cell axon tracing analysis." Front Neuroanat 9: 59.

Arbuthnott, G. W. and J. Wickens (2007). "Space, time and dopamine." Trends Neurosci 30(2): 62-69.

Arnsten, A. F., J. X. Cai, B. L. Murphy and P. S. Goldman-Rakic (1994). "Dopamine D1 receptor mechanisms in the cognitive performance of young adult and aged monkeys." Psychopharmacology (Berl) 116(2): 143-151.

Barthas, F. and A. C. Kwan (2017). "Secondary motor cortex: where ‘sensory’meets ‘motor’in the rodent frontal cortex." Trends in neurosciences 40(3): 181-193.

Berger, B., P. Gaspar and C. Verney (1991). "Dopaminergic innervation of the cerebral cortex: unexpected differences between rodents and primates." Trends Neurosci 14(1): 21-27.

Caballero, A. and K. Y. Tseng (2016). "GABAergic Function as a Limiting Factor for Prefrontal Maturation during Adolescence." Trends Neurosci 39(7): 441-448.

Ellwood, I. T., T. Patel, V. Wadia, A. T. Lee, A. T. Liptak, K. J. Bender and V. S. Sohal (2017). "Tonic or Phasic Stimulation of Dopaminergic Projections to Prefrontal Cortex Causes Mice to Maintain or Deviate from Previously Learned Behavioral Strategies." J Neurosci 37(35): 8315-8329.

Floresco, S. B. (2013). "Prefrontal dopamine and behavioral flexibility: shifting from an "inverted-U" toward a family of functions." Front Neurosci 7: 62.

Fromer, M., A. J. Pocklington, D. H. Kavanagh, H. J. Williams, S. Dwyer, P. Gormley, L. Georgieva, E. Rees, P. Palta, D. M. Ruderfer, N. Carrera, I. Humphreys, J. S. Johnson, P. Roussos, D. D. Barker, E. Banks, V. Milanova, S. G. Grant, E. Hannon, S. A. Rose, K. Chambert, M. Mahajan, E. M. Scolnick, J. L. Moran, G. Kirov, A. Palotie, S. A. McCarroll, P. Holmans, P. Sklar, M. J. Owen, S. M. Purcell and M. C. O'Donovan (2014). "De novo mutations in schizophrenia implicate synaptic networks." Nature 506(7487): 179-184.

Gireesh, E. D. and D. Plenz (2008). "Neuronal avalanches organize as nested theta- and beta/gamma-oscillations during development of cortical layer 2/3." Proc Natl Acad Sci U S A 105(21): 7576-7581.

Goldman-Rakic, P. S., C. Leranth, S. M. Williams, N. Mons and M. Geffard (1989). "Dopamine synaptic complex with pyramidal neurons in primate cerebral cortex." Proc Natl Acad Sci U S A 86(22): 9015-9019.

Gunaydin, L. A., L. Grosenick, J. C. Finkelstein, I. V. Kauvar, L. E. Fenno, A. Adhikari, S. Lammel, J. J. Mirzabekov, R. D. Airan, K. A. Zalocusky, K. M. Tye, P. Anikeeva, R. C. Malenka and K. Deisseroth (2014). "Natural neural projection dynamics underlying social behavior." Cell 157(7): 1535-1551.

Hoops, D. and C. Flores (2017). "Making Dopamine Connections in Adolescence." Trends Neurosci 40(12): 709-719.

Kalsbeek, A., P. Voorn, R. M. Buijs, C. W. Pool and H. B. Uylings (1988). "Development of the dopaminergic innervation in the prefrontal cortex of the rat." J Comp Neurol 269(1): 58-72.

Lammel, S., A. Hetzel, O. Hackel, I. Jones, B. Liss and J. Roeper (2008). "Unique properties of mesoprefrontal neurons within a dual mesocorticolimbic dopamine system." Neuron 57(5): 760-773.

Lammel, S., A. Hetzel, O. Haeckel, I. Jones, B. Liss and J. Roeper (2008). "Unique properties of mesoprefrontal neurons within a dual mesocorticolimbic dopamine system." Neuron 57(5): 760-773.

Lammel, S., B. K. Lim, C. Ran, K. W. Huang, M. J. Betley, K. M. Tye, K. Deisseroth and R. C. Malenka (2012). "Input-specific control of reward and aversion in the ventral tegmental area." Nature 491(7423): 212-217.

Lammel, S., E. E. Steinberg, C. Foldy, N. R. Wall, K. Beier, L. Luo and R. C. Malenka (2015). "Diversity of transgenic mouse models for selective targeting of midbrain dopamine neurons." Neuron 85(2): 429-438.

Larsen, B. and B. Luna (2018). "Adolescence as a neurobiological critical period for the development of higher-order cognition." Neurosci Biobehav Rev 94: 179-195.

Lewis, B. L. and P. O'Donnell (2000). "Ventral tegmental area afferents to the prefrontal cortex maintain membrane potential 'up' states in pyramidal neurons via D(1) dopamine receptors." Cereb Cortex 10(12): 1168-1175.

Li, L., B. Tasic, K. D. Micheva, V. M. Ivanov, M. L. Spletter, S. J. Smith and L. Luo (2010). "Visualizing the distribution of synapses from individual neurons in the mouse brain." PLoS One 5(7): e11503.

Li, X., J. Qi, T. Yamaguchi, H. L. Wang and M. Morales (2013). "Heterogeneous composition of dopamine neurons of the rat A10 region: molecular evidence for diverse signaling properties." Brain Struct Funct 218(5): 1159-1176.

Liu, C., P. Goel and P. S. Kaeser (2021). "Spatial and temporal scales of dopamine transmission." Nat Rev Neurosci 22(6): 345-358.

Lohani, S., A. K. Martig, K. Deisseroth, I. B. Witten and B. Moghaddam (2019). "Dopamine Modulation of Prefrontal Cortex Activity Is Manifold and Operates at Multiple Temporal and Spatial Scales." Cell Rep 27(1): 99-114 e116.

Manago, F., M. Mereu, S. Mastwal, R. Mastrogiacomo, D. Scheggia, M. Emanuele, M. A. De Luca, D. R. Weinberger, K. H. Wang and F. Papaleo (2016). "Genetic Disruption of Arc/Arg3.1 in Mice Causes Alterations in Dopamine and Neurobehavioral Phenotypes Related to Schizophrenia." Cell Rep 16(8): 2116-2128.

Mastwal, S., Y. Ye, M. Ren, D. V. Jimenez, K. Martinowich, C. R. Gerfen and K. H. Wang (2014). "Phasic dopamine neuron activity elicits unique mesofrontal plasticity in adolescence." J Neurosci 34(29): 9484-9496.

Morales, M. and E. B. Margolis (2017). "Ventral tegmental area: cellular heterogeneity, connectivity and behaviour." Nat Rev Neurosci 18(2): 73-85.

Mukherjee, A., F. Carvalho, S. Eliez and P. Caroni (2019). "Long-Lasting Rescue of Network and Cognitive Dysfunction in a Genetic Schizophrenia Model." Cell 178(6): 1387-1402 e1314.Murty, V. P., F. Calabro and B. Luna (2016). "The role of experience in adolescent cognitive development: Integration of executive, memory, and mesolimbic systems." Neurosci Biobehav Rev 70: 46-58.

Naneix, F., A. R. Marchand, G. Di Scala, J. R. Pape and E. Coutureau (2012). "Parallel maturation of goal-directed behavior and dopaminergic systems during adolescence." J Neurosci 32(46): 16223-16232.

Niwa, M., H. Jaaro-Peled, S. Tankou, S. Seshadri, T. Hikida, Y. Matsumoto, N. G. Cascella, S. Kano, N. Ozaki, T. Nabeshima and A. Sawa (2013). "Adolescent stress-induced epigenetic control of dopaminergic neurons via glucocorticoids." Science 339(6117): 335-339.

Niwa, M., A. Kamiya, R. Murai, K. Kubo, A. J. Gruber, K. Tomita, L. Lu, S. Tomisato, H. Jaaro-Peled, S. Seshadri, H. Hiyama, B. Huang, K. Kohda, Y. Noda, P. O'Donnell, K. Nakajima, A. Sawa and T. Nabeshima (2010). "Knockdown of DISC1 by in utero gene transfer disturbs postnatal dopaminergic maturation in the frontal cortex and leads to adult behavioral deficits." Neuron 65(4): 480-489.

O'Donnell, P. (2010). "Adolescent maturation of cortical dopamine." Neurotox Res 18(3-4): 306-312.

Oh, S. W., J. A. Harris, L. Ng, B. Winslow, N. Cain, S. Mihalas, Q. Wang, C. Lau, L. Kuan, A. M. Henry, M. T. Mortrud, B. Ouellette, T. N. Nguyen, S. A. Sorensen, C. R. Slaughterbeck, W. Wakeman, Y. Li, D. Feng, A. Ho, E. Nicholas, K. E. Hirokawa, P. Bohn, K. M. Joines, H. Peng, M. J. Hawrylycz, J. W. Phillips, J. G. Hohmann, P. Wohnoutka, C. R. Gerfen, C. Koch, A. Bernard, C. Dang, A. R. Jones and H. Zeng (2014). "A mesoscale connectome of the mouse brain." Nature 508(7495): 207-214.

Papathanou, M., S. Dumas, H. Pettersson, L. Olson and A. Wallen-Mackenzie (2019). "Off-Target Effects in Transgenic Mice: Characterization of Dopamine Transporter (DAT)-Cre Transgenic Mouse Lines Exposes Multiple Non-Dopaminergic Neuronal Clusters Available for Selective Targeting within Limbic Neurocircuitry." eNeuro 6(5).

Patriarchi, T., J. R. Cho, K. Merten, M. W. Howe, A. Marley, W. H. Xiong, R. W. Folk, G. J. Broussard, R. Liang, M. J. Jang, H. Zhong, D. Dombeck, M. von Zastrow, A. Nimmerjahn, V. Gradinaru, J. T. Williams and L. Tian (2018). "Ultrafast neuronal imaging of dopamine dynamics with designed genetically encoded sensors." Science 360(6396): 1420-+.

Porter, L. L., E. Rizzo and J. P. Hornung (1999). "Dopamine affects parvalbumin expression during cortical development in vitro." J Neurosci 19(20): 8990-9003.

Purcell, S. M., J. L. Moran, M. Fromer, D. Ruderfer, N. Solovieff, P. Roussos, C. O'Dushlaine, K. Chambert, S. E. Bergen, A. Kahler, L. Duncan, E. Stahl, G. Genovese, E. Fernandez, M. O. Collins, N. H. Komiyama, J. S. Choudhary, P. K. Magnusson, E. Banks, K. Shakir, K. Garimella, T. Fennell, M. DePristo, S. G. Grant, S. J. Haggarty, S. Gabriel, E. M. Scolnick, E. S. Lander, C. M. Hultman, P. F. Sullivan, S. A. McCarroll and P. Sklar (2014). "A polygenic burden of rare disruptive mutations in schizophrenia." Nature 506(7487): 185-190.

Robbins, T. W. (2000). "Chemical neuromodulation of frontal-executive functions in humans and other animals." Exp Brain Res 133(1): 130-138.

Rolls, E. T., M. Loh, G. Deco and G. Winterer (2008). "Computational models of schizophrenia and dopamine modulation in the prefrontal cortex." Nat Rev Neurosci 9(9): 696-709.

Seamans, J. K. and C. R. Yang (2004). "The principal features and mechanisms of dopamine modulation in the prefrontal cortex." Prog Neurobiol 74(1): 1-58.

Sesack, S. R., V. A. Hawrylak, C. Matus, M. A. Guido and A. I. Levey (1998). "Dopamine axon varicosities in the prelimbic division of the rat prefrontal cortex exhibit sparse immunoreactivity for the dopamine transporter." J Neurosci 18(7): 2697-2708.

Soden, M. E., S. M. Miller, L. M. Burgeno, P. E. M. Phillips, T. S. Hnasko and L. S. Zweifel (2016). "Genetic Isolation of Hypothalamic Neurons that Regulate Context-Specific Male Social Behavior." Cell Rep 16(2): 304-313.

Spear, L. (2000). "Modeling adolescent development and alcohol use in animals." Alcohol Res Health 24(2): 115-123.

Stagkourakis, S., G. Spigolon, P. Williams, J. Protzmann, G. Fisone and C. Broberger (2018). "A neural network for intermale aggression to establish social hierarchy." Nat Neurosci 21(6): 834-842.Sul, J. H., S. Jo, D. Lee and M. W. Jung (2011). "Role of rodent secondary motor cortex in value-based action selection." Nat Neurosci 14(9): 1202-1208.

Sul, J. H., H. Kim, N. Huh, D. Lee and M. W. Jung (2010). "Distinct roles of rodent orbitofrontal and medial prefrontal cortex in decision making." Neuron 66(3): 449-460.

Sulzer, D., S. J. Cragg and M. E. Rice (2016). "Striatal dopamine neurotransmission: regulation of release and uptake." Basal Ganglia 6(3): 123-148.

Tseng, K. Y. and P. O'Donnell (2007). "Dopamine modulation of prefrontal cortical interneurons changes during adolescence." Cereb Cortex 17(5): 1235-1240.

Vander Weele, C. M., C. A. Siciliano, G. A. Matthews, P. Namburi, E. M. Izadmehr, I. C. Espinel, E. H. Nieh, E. H. S. Schut, N. Padilla-Coreano, A. Burgos-Robles, C. J. Chang, E. Y. Kimchi, A. Beyeler, R. Wichmann, C. P. Wildes and K. M. Tye (2018). "Dopamine enhances signal-to-noise ratio in cortical-brainstem encoding of aversive stimuli." Nature 563(7731): 397-401.

Vijayraghavan, S., M. Wang, S. G. Birnbaum, G. V. Williams and A. F. Arnsten (2007). "Inverted-U dopamine D1 receptor actions on prefrontal neurons engaged in working memory." Nat Neurosci 10(3): 376-384.

Wen, Z., H. N. Nguyen, Z. Guo, M. A. Lalli, X. Wang, Y. Su, N. S. Kim, K. J. Yoon, J. Shin, C. Zhang, G. Makri, D. Nauen, H. Yu, E. Guzman, C. H. Chiang, N. Yoritomo, K. Kaibuchi, J. Zou, K. M. Christian, L. Cheng, C. A. Ross, R. L. Margolis, G. Chen, K. S. Kosik, H. Song and G. L. Ming (2014). "Synaptic dysregulation in a human iPS cell model of mental disorders." Nature 515(7527): 414-418.

Wood, J., Y. Kim and B. Moghaddam (2012). "Disruption of prefrontal cortex large scale neuronal activity by different classes of psychotomimetic drugs." J Neurosci 32(9): 3022-3031.

Ye, Y., S. Mastwal, V. Y. Cao, M. Ren, Q. Liu, W. Zhang, A. G. Elkahloun and K. H. Wang (2017). "Dopamine is Required for Activity-Dependent Amplification of Arc mRNA in Developing Postnatal Frontal Cortex." Cereb Cortex 27(7): 3600-3608.